# Secretary and Online Matching Problems with Machine Learned Advice

**Antonios Antoniadis**
Universität zu Köln
Köln, Germany
antoniadis@cs.uni-koeln.de

**Themis Gouleakis**
Max Planck Institute for Informatics
Saarland Informatics Campus
Saarbrücken, Germany
tgouleak@mpi-inf.mpg.de

**Pieter Kleer**
Max Planck Institute for Informatics
Saarland Informatics Campus
Saarbrücken, Germany
pkleer@mpi-inf.mpg.de

**Pavel Kolev**
Max Planck Institute for Informatics
Saarland Informatics Campus
Saarbrücken, Germany
pkolev@mpi-inf.mpg.de

## Abstract

The classical analysis of online algorithms, due to its worst-case nature, can be quite pessimistic when the input instance at hand is far from worst-case. Often this is not an issue with machine learning approaches, which shine in exploiting patterns in past inputs in order to predict the future. However, such predictions, although usually accurate, can be arbitrarily poor. Inspired by a recent line of work, we augment three well-known online settings with machine learned predictions about the future, and develop algorithms that take them into account. In particular, we study the following online selection problems: (i) the classical secretary problem, (ii) online bipartite matching and (iii) the graphic matroid secretary problem. Our algorithms still come with a worst-case performance guarantee in the case that predictions are subpar while obtaining an improved competitive ratio (over the best-known classical online algorithm for each problem) when the predictions are sufficiently accurate. For each algorithm, we establish a trade-off between the competitive ratios obtained in the two respective cases.

## 1  Introduction

There has been enormous progress in the field of machine learning in the last decade, which has affected a variety of other areas as well. One of these areas is the design of online algorithms. Traditionally, the analysis of such algorithms involves worst-case guarantees, which can often be quite pessimistic. It is conceivable though, that having prior knowledge regarding the online input (obtained using machine learning) could potentially improve those guarantees significantly.

In this work, we consider various online selection algorithms augmented with so-called *machine learned advice*. In particular, we consider secretary and online bipartite matching problems. The high-level idea is to incorporate some form of *predictions* in an existing online algorithm in order to get the best of two worlds: (i) provably improve the algorithm's performance guarantee in the case that predictions are sufficiently good, while (ii) losing only a constant factor of the algorithm's existing worst-case performance guarantee, when the predictions are subpar. Improving the performance of classical online algorithms with the help of machine learned predictions is a relatively new area that has gained a lot of attention in the last couple of years [24, 21, 36, 27, 30, 34, 17, 33, 32].

We motivate the idea of incorporating such machine-learned advice, in the class of problems studied in this work, by illustrating a simple real-world problem. Consider the following setting for selling a laptop on an online platform.[1] Potential buyers arrive one by one, say, in a uniformly random order, and report a price that they are willing to pay for the laptop. Whenever a buyer arrives, we have to irrevocably decide if we want to sell at the given price, or wait for a better offer. Based on historical data, e.g., regarding previous online sales of laptops with similar specs, the online platform might suggest a (machine learned) prediction for the maximum price that some buyer is likely to offer.

How can we exploit this information in our decision process? One problem that arises here is that we do not have any formal guarantees for how accurate the machine-learned advice is for any particular instance. For example, suppose we get a prediction of 900 dollars as the maximum price that some buyer will likely offer. One extreme policy is to blindly trust this prediction and wait for the first buyer to come along that offers a price sufficiently close to 900 dollars. If this prediction is indeed accurate, this policy has an almost perfect performance guarantee, in the sense that we will sell to the (almost) highest bidder. However, if the best offer is only, say, 500 dollars, we will never sell to this buyer (unless this offer arrives last), since the advice is to wait for a better offer to come along. In particular, the performance guarantee of this selling policy depends on the *prediction error* (400 dollars in this case) which can become arbitrarily large. The other extreme policy is to completely ignore the prediction of 900 dollars and just run the classical secretary algorithm: Observe a $1/e$-fraction of the buyers, and then sell to the first buyer that arrives afterwards, who offers a price higher than the best offer seen in the initially observed fraction. This yields, in expectation, a selling price of at least $1/e$ times the highest offer [28, 11].

Can we somehow combine the preceding two extreme selling-policies, so that we get a performance guarantee strictly better than that of $1/e$ in the case where the prediction for the highest offer is not too far off, while not loosing too much over the guarantee of $1/e$ otherwise? Note that (even partially) trusting poor predictions often comes at a price, and thus obtaining a competitive ratio worse than $1/e$ seems inevitable in this case. We show that there is in fact a trade-off between the competitive ratio that we can achieve when the prediction is accurate and the one we obtain when the prediction error turns out to be large.

## 1.1  Our models and contributions

We show how one can incorporate predictions in various online selection algorithms for problems that generalize the classical secretary problem. The overall goal is to include *as little predictive information as possible* into the algorithm, while still obtaining improvements in the case that the information is accurate. Our results are parameterized by (among other parameters) the so-called prediction error $\eta$ that measures the quality of the given predictions. We note that for all the considered problems, one cannot hope for an algorithm with a performance guarantee better than $1/e$ in the corresponding settings without predictions, as this bound is known to be optimal for the classical secretary problem (and also applies to the other problems) [28, 11]. Our goal is to design algorithms that improve upon the $1/e$ worst-case competitive ratio in the case where the prediction error is sufficiently small, and otherwise (when the prediction error is large) never lose more than a constant (multiplicative) factor over the worst-case competitive ratio. More specifically: We start by introducing a meta-result in Theorem 1.1, that applies to all of the problems, before introducing each of them individually along with the specific corresponding result.

**Theorem 1.1** (Meta Result). *For any $\lambda \geq 0$, there is a polynomial time deterministic algorithm that incorporates the predictions $p^*$ such that for some constants $0 < \alpha, \beta < 1$ it is*
*(i) $\alpha$-competitive with $\alpha > 1/e$, when the prediction error is sufficiently small; and*
*(ii) $\beta$-competitive with $\beta < 1/e$, independently of the prediction error.*

We note that there is a correlation between the constants $\alpha$ and $\beta$, which can be intuitively described as follows: The more one is willing to give up in the worst-case guarantee, i.e. the more confidence we have in the predictions, the better the competitive ratio becomes in the case where the predictions are sufficiently accurate. Each of our algorithms takes as input a parameter $\lambda \geq 0$ which quantifies this level of confidence (small $\lambda$ implies more confidence and vice-versa). We now take a closer look at our contributions for each specific problem:

**Secretary problem.** In the secretary problem there is a set $\{1, \ldots, n\}$ of secretaries, each with a value $v_i \geq 0$ for $i \in \{1, \ldots, n\}$, that arrive in a *uniformly random order*. Whenever a secretary arrives, we have to irrevocably decide whether we want to hire that person. If we decide to hire a secretary, we automatically reject all subsequent candidates. The goal is to select the secretary with the highest value. We assume without loss of generality that all values $v_i$ are distinct. This can be done by introducing a suitable tie-breaking rule if necessary. For details, see Section 3.

In order to illustrate our ideas and techniques, we start by augmenting the classical secretary problem[2] with predictions. Here, we are given a prediction $p^*$ for the maximum value among all arriving secretaries.[3] The prediction error is then defined as $\eta = |p^* - v^*|$, where $v^*$ is the true maximum value among all secretaries. We emphasize that the algorithm is not aware of the prediction error $\eta$, and this parameter is only used to analyze the algorithm's performance guarantee. We show the following theorem:

**Theorem 1.2.** *For any $\lambda \geq 0$ and $c > 1$, there is a deterministic algorithm for the (value-maximization) secretary problem that is asymptotically $g_{c,\lambda}(\eta)$-competitive in expectation, where*

$$g_{c,\lambda}(\eta) = \left\{ \begin{array}{ll} \max \left\{ \frac{1}{ce}, \left[ f(c) \left( \max \left\{ 1 - \frac{\lambda+\eta}{OPT}, 0 \right\} \right) \right] \right\} & \text{if } 0 \leq \eta < \lambda \\ \frac{1}{ce} & \text{if } \eta \geq \lambda \end{array} \right\},$$

*and the function $f(c)$ is given in terms of the two branches $W_0$ and $W_{-1}$ of the Lambert $W$-function[4] and reads $f(c) = \exp\{W_0(-1/(ce))\} - \exp\{W_{-1}(-1/(ce))\}$.*

We note that $\lambda$ and $c$ are independent parameters that provide the most general description of the competitive ratio. Here $\lambda$ is our confidence in the predictions and $c$ describes how much we are willing to lose in the worst case. Although these parameters can be set independently, some combinations of them are not very sensible, as one might not get an improved performance guarantee, even when the prediction error is small (for instance, if $c = 1$, i.e., we are not willing to lose anything in the worst case, then it is not helpful to consider the prediction at all). To illustrate the influence of these parameters on the competitive ratio, in Figure 1, we plot various combinations of the input parameters $c, \lambda$ and $p^*$ of Algorithm 1 (in Section 3), assuming that $\eta = 0$. In this case $p^* = \text{OPT}$ and the competitive ratio simplifies to

$$g_{c,\lambda}(0) = \max \left\{ \frac{1}{ce}, f(c) \cdot \max \left\{ 1 - \frac{\lambda}{p^*}, 0 \right\} \right\}.$$

We therefore choose the axes of Figure 1 to be $\lambda/p^*$ and $c$.

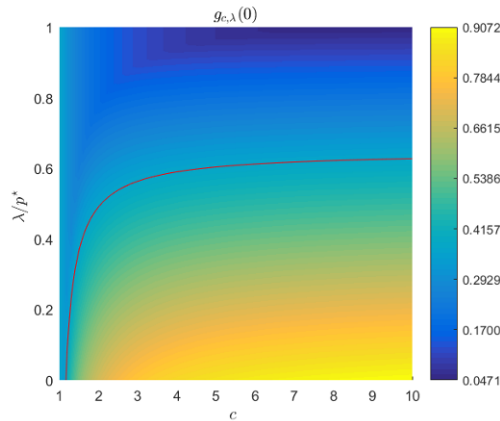

Figure 1: The red curve shows the optimal competitive ratio without predictions, i.e., $g_{c,\lambda}(0) = 1/e$. Our algorithm achieves an improved competitive ratio $g_{c,\lambda}(0) > 1/e$ in the area below this curve, and a worse competitive ratio $g_{c,\lambda}(0) < 1/e$ in the area above it.

Furthermore, as one does not know the prediction error $\eta$, there is no way in choosing these parameters optimally, since different prediction errors require different settings of $\lambda$ and $c$.

To get an impression of the statement in Theorem 1.2, if we have, for example, $\eta + \lambda = \frac{1}{10}$OPT, then we start improving over $1/e$ for $c \geq 1.185$. Moreover, if one believes that the prediction error is low, one should set $c$ very high (hence approaching a 1-competitive algorithm in case the predictions are close to perfect). Note also, that the bound obtained in Theorem 1.2 has a discontinuity at $\eta = \lambda$. This can be easily smoothed out by selecting $\lambda$ according to some distribution, which now represents our confidence in the prediction $p^*$. The competitive ratio will start to drop earlier in this case, and will continuously reach $1/(ce)$. This bound is tight for any fixed $c > 1$ when $\eta = \lambda = 0$. We illustrate how the competitive ratio changes as a function of $\eta$ in the supplementary material to this paper.

**Online bipartite matching with vertex arrivals.** We study the online bipartite matching problem in which the set of nodes $L$ of a bipartite graph $G = (L \cup R, E)$, with $|L| = n$ and $|R| = m$, arrives *online* in a uniformly random order [23, 20]. Upon arrival, a node reveals the edge-weights to its neighbors in $R$. We have to irrevocably decide if we want to match up the arrived online node with one of its (currently unmatched) neighbors in $R$. Kesselheim et al. [20] gave a tight $1/e$-competitive deterministic algorithm for this setting that significantly generalizes the same guarantee for the classical secretary algorithm [28, 11].

The prediction that we consider in this setting is a vector of values $p^* = (p_1^*, \ldots, p_m^*)$ that predicts the edge-weights adjacent to the nodes $r \in R$ in some fixed optimal (offline) bipartite matching. That is, the prediction $p^*$ indicates the existence of a fixed optimal bipartite matching in which each node $r \in R$ is adjacent to an edge with weight $p_r^*$. The prediction error is then the maximum prediction error taken over all nodes in $r \in R$ and minimized over all optimal matchings. This generalizes the prediction used for the classical secretary problem. This type of predictions closely corresponds to the *vertex-weighted online bipartite matching problem* [1], which is discussed in the supplementary material to this paper. More formally, we prove the following theorem about the online bipartite matching problem with vertex arrivals:

**Theorem 1.3.** *For any $\lambda \geq 0$ and $c > d \geq 1$, there is a deterministic algorithm for the online bipartite matching problem with uniformly random arrivals that is asymptotically $g_{c,d,\lambda}(\eta)$-competitive in expectation, where*

$$g_{c,d,\lambda}(\eta) = \left\{ \begin{array}{ll} \max\left\{ \frac{1}{c}\ln(\frac{c}{d}), \left[\frac{d-1}{2c}\left(\max\left\{1 - \frac{(\lambda+\eta)|\psi|}{OPT}, 0\right\}\right)\right]\right\} & \text{if } 0 \leq \eta < \lambda, \\ \frac{1}{c}\ln(\frac{c}{d}) & \text{if } \eta \geq \lambda. \end{array} \right\},$$

*and $|\psi|$ is the cardinality of an optimal (offline) matching $\psi$ of the instance.*

the bound in $g_{c,d,\lambda}$... If $\lambda$ is small, we roughly obtain a bound of $(d-1)/2c$ in case $\eta$ is small as well, and a bound of $\ln(c/d)/c$ if $\eta$ is large. Moreover, when $c/d \to 1$, we approach a bound of $1/2$ in case $\eta$ is small, whereas the worst-case guarantee $\ln(c/d)/c$ for large $\eta$ then increases. In other words, although the predictions provide very little information about the optimal solution, we are still able to obtain improved theoretical guarantees in the case where the predictions are sufficiently accurate. In the online bipartite matching setting with predictions for the nodes in $R$, we can essentially get close to a $1/2$-approximation – which is best possible – assuming the predictions are close to perfect. This follows from the fact that in this case we obtain the so-called *vertex-weighted online bipartite matching problem* for which there is a deterministic $1/2$-competitive algorithm, and no algorithm can do better [1]. Our algorithm "converges" to that in [1] when the predictions get close to perfect.

**Graphic matroid secretary problem.** We also augment the graphic matroid secretary problem with predictions. In this problem, the edges of a given undirected graph $G = (V, E)$, with $|V| = n$ and $|E| = m$, arrive in a uniformly random order. The goal is to select a subset of edges of maximum weight under the constraint that this subset is a forest. That is, it is not allowed to select a subset of edges that form a cycle in $G$. The best known algorithm for this online problem is a (randomized) $1/4$-competitive algorithm by Soto, Turkieltaub and Verdugo [37]. Their algorithm proceeds by first selecting no elements from a prefix of the sequence of elements with randomly chosen size, followed by selecting an element if and only if it belongs to a (canonically computed) offline optimal solution, and can be added to the set of elements currently selected online. This is inspired by the algorithm of Kesselheim et al. [20] for online bipartite matching.

As a result of possible independent interest, we show that there exists a *deterministic* $(1/4 - o(1))$-competitive algorithm for the graphic matroid secretary problem, which can roughly be seen as a

deterministic version of the algorithm of Soto et al. [37]. Alternatively, our algorithm can be seen as a variation on the (deterministic) algorithm of Kesselheim et al. [20] for the case of online bipartite matching (in combination with an idea introduced in [4]).

The prediction that we consider here is a vector of values $p = (p_1^*, \ldots, p_n^*)$ where $p_i^*$ predicts the maximum edge-weight that node $i \in V$ is adjacent to, in the graph $G$. This is equivalent to saying that $p_i^*$ is the maximum edge-weight adjacent to node $i \in V$ in a given optimal spanning tree (we assume that $G$ is connected for sake of simplicity), which is, in a sense, in line with the predictions used for the preceding problems. (We note that the predictions model the optimal spanning tree in the case when all edge-weights are pairwise distinct. Otherwise, there can be many (offline) optimal spanning trees, and thus the predictions do not encode a unique optimal spanning tree. We intentionally chose not to use predictions regarding which edges are part of an optimal solution, as in our opinion, such an assumption would be too strong.) The prediction error is also defined similarly. We show the following:

**Theorem 1.4.** *For any $\lambda \geq 0$ and $c > d \geq 1$, there is a deterministic algorithm for the graphic matroid secretary problem that is asymptotically $g_{c,d,\lambda}(\eta)$-competitive in expectation, where*

$$
g_{c,d,\lambda}(\eta) = \begin{cases} \max\left\{ \frac{d-1}{c^2}, \frac{1}{2}\left(\frac{1}{d} - \frac{1}{c}\right)\left(1 - \frac{2(\lambda+\eta)|V|}{OPT}\right) \right\} & \text{if } 0 \leq \eta < \lambda, \\ \frac{d-1}{c^2} & \text{if } \eta \geq \lambda. \end{cases}
$$

If $\lambda$ is small, we roughly obtain a bound of $(1/d - 1/c)/2$ in case $\eta$ is small, and a bound of $(d-1)/c^2$ if $\eta$ is large. Roughly speaking, if $d \to 1$ and $c \to \infty$ we approach a bound of $1/2$ when the predictions are good, whereas the bound of $(d-1)/c^2$ becomes arbitrarily bad. For this problem, we get close to a $1/2$-approximation in the case where the predictions (the maximum edge-weights adjacent to the nodes in the graph) get close to perfect. However this is probably not tight. We suspect that, when given perfect predictions, it is possible to obtain an algorithm with a better approximation guarantee. This is an interesting open problem.

**Discussion and Extensions.** For each of the preceding three problems, we give a *deterministic* algorithm. In particular, we show that the canonical approaches for the secretary problem [28, 11] and the online bipartite matching problem [20] can be naturally augmented with predictions. We also show how to adapt our novel deterministic algorithm for the graphic matroid secretary problem. Furthermore, we comment on randomized approaches for each respective problem in the supplementary material.

The high-level overview of our approach is as follows. We split the sequence of uniformly random arrivals in three phases. In the first phase, we observe a fraction of the input without selecting anything. In the remaining two phases, we run two extreme policies, which either exploit the predictions or ignore them completely. Although, each of the aforementioned extreme policies can be analyzed individually, using existing techniques, it is a non-trivial task to show that when combined they do not obstruct each other too much. The execution order of these policies is also crucial for the analysis.

As an additional result, we show that the online bipartite matching algorithm, can be turned into a truthful mechanism in the case of the so-called *single-value unit-demand domains*. We show that the algorithm of Kesselheim et al. [20] can be turned into a truthful mechanism in the special case of *uniform edge-weights*, where for every fixed online node in $L$, there is a common weight on all edges adjacent to it. In addition, we show that truthfulness can be preserved when predictions are included in the algorithm. We note that Reiffenhäuser [35] recently gave a truthful mechanism for the general online bipartite matching setting (without uniform edge-weights). It would be interesting to see if her algorithm can be augmented with predictions as well.

**Remark 1.5.** *In the statements of Theorem 1.2, 1.3 and 1.4 it is assumed that the set of objects arriving online (either vertices or edges) is asymptotically large. We hide $o(1)$-terms at certain places for the sake of readability.*

## 1.2 Related work

This subsection consists of three parts. First we discuss relevant approximation algorithms for the matroid secretary problem without any form of prior information, then we consider models that incorporate additional information, such as the area of prophet inequalities. Finally, we give a short overview of related problems that have been analyzed with the inclusion of machine learned advice following the frameworks in [30, 34], which we study here as well.

**Approximation algorithms for the matroid secretary problem.** The classical secretary problem was originally introduced by Gardner [15], and solved by Lindley [28] and Dynkin [11], who gave $1/e$-competitive algorithms. Babaioff et al. [4] introduced the matroid secretary problem, a considerable generalization of the classical secretary problem, where the goal is to select a set of secretaries with maximum total value under a matroid constraint for the set of feasible secretaries. They provided an $O(1/\log(r))$-competitive algorithm for this problem, where $r$ is the rank of the underlying matroid. Lachish [26] later gave an $O(1/\log\log(r))$-competitive algorithm, and a simplified algorithm with the same guarantee was given by Feldman, Svensson and Zenklusen [13]. It is still a major open problem if there exists a constant-competitive algorithm for the matroid secretary problem. Nevertheless, many constant-competitive algorithms are known for special classes of matroids, and we mention those relevant to the results in this work (see, e.g., [4, 37] for further related work). Babaioff et al. [4] provided a $1/16$-competitive algorithm for the case of transversal matroids, which was later improved to a $1/8$-competitive algorithm by Dimitrov and Plaxton [9]. Korula and Pál [23] provided the first constant competitive algorithm for the online bipartite matching problem, of which the transversal matroid secretary problem is a special case. In particular, they gave a $1/8$-approximation. Kesselheim et al. [20] provided a $1/e$-competitive algorithm, which is best possible, as discussed above. For the graphic matroid secretary problem, there was a series of approximation algorithms [4, 3, 23]. The state of the art is a randomized $1/4$-competitive algorithm by Soto et al. [37]. All algorithms run in polynomial time.

**Other models, extensions and variations.** There is a vast literature on online selection algorithms for problems similar to the (matroid) secretary problem. Here we discuss some recent directions and other models incorporating some form of prior information. The most important assumption in the secretary model that we consider is the fact that elements arrive in a uniformly random order. If the elements arrive in an adversarial order, there is not much one can achieve: There is a trivial randomized algorithm that selects every element with probability $1/n$, yielding a $1/n$-competitive algorithm; deterministically no finite competitive algorithm is possible with a guarantee independent of the values of the elements. There has been a recent interest in studying *intermediate* arrival models that are not completely adversarial, nor uniformly random. Kesselheim, Kleinberg and Niazadeh [19] study non-uniform arrival orderings under which (asymptotically) one can still obtain a $1/e$-competitive algorithm for the secretary problem. Bradac et al. [5] consider the so-called Byzantine secretary model in which some elements arrive uniformly at random, but where an adversary controls a set of elements that can be inserted in the ordering of the uniform elements in an adversarial manner. See also the very recent work of Garg et al. [16] for a conceptually similar model. In a slightly different setting, Kaplan et al. [18] consider a secretary problem with the assumption that the algorithm has access to a random sample of the adversarial distribution ahead of time. For this setting they provide an algorithm with almost tight competitive-ratio for small sample-sizes. Furthermore, there is also a vast literature on so-called *prophet inequalities*. In the basic model, the elements arrive in an adversarial order, but there is a prior distributional information given for the values of the elements $\{1, \ldots, n\}$. That is, one is given probability distributions $X_1, \ldots, X_n$ from which the values of the elements are drawn. Upon arrival of an element $e$, its value drawn according to $X_e$ is revealed and an irrevocable decision is made whether to select this element or not. Note that the available distributional information can be used to decide on whether to select an element. The goal is to maximize the expected value, taken over all prior distributions, of the selected element. For surveys on recent developments, refer to [29, 7]. Here we discuss some classical results and recent related works. Krengel, Sucheston and Garling [25] show that there is an optimal $1/2$-competitive algorithm for this problem. Kleinberg and Weinberg [22] gave a significant generalization of this result to matroid prophet inequalities, where multiple elements can be selected subject to a matroid feasibility constraint (an analogue of the matroid secretary problem). There is also a growing interest in the prophet secretary problem [12], in which the elements arrive uniformly random (as in the secretary problem); see also [7]. Recently, settings with more limited prior information gained a lot of interest. These works address the quite strong assumption of knowing all element-wise prior distributions. Azar, Kleinberg and Weinberg [2] study the setting in which one has only access to one sample from every distribution, as opposed to the whole distribution; see also [38]. Correa et al. [6] study this problem under the assumption that all elements are identically distributed. Recently, an extension of this setting was considered by Correa et al. [8]. Furthermore, Dütting and Kesselheim [10] consider prophet inequalities with inaccurate prior distributions $\tilde{X}_1, \ldots, \tilde{X}_n$ (while the true distributions $X_1, \ldots, X_n$ are unknown) and study the extent to which existing algorithms are robust against inaccurate distributions.

Although our setting also assumes additional information about the input instance, there are major differences. Mainly, we are interested in including a minimal amount of predictive information about an optimal (offline) solution, which yields a quantitative improvement in the case where the prediction is sufficiently accurate. This is a much weaker assumption than having a priori all element-wise (possibly inaccurate) probability distributions. Furthermore, our setting does not assume that the predictive information necessarily comes from a distribution (which is then used to measure the expected performance of an algorithm), but can be obtained in a more general fashion from historical data (using, e.g., statistical or machine learning techniques). Finally, and in contrast to other settings, the information received in our setting can be inaccurate (and this is non-trivial do deal with).

**Other problems with black-box machine learned advice**   Although online algorithms with machine learned advice is a relatively new area, there has already been a number of interesting results. Most of the following results are analyzed by means of *consistency* (competitive-ratio in the case of perfect predictions) and *robustness* (worst-case competitive-ratio regardless of prediction quality), but the precise formal definitions of consistency and robustness slightly differ in each work [30, 34].[5] Our results can also be interpreted within this framework, but for the sake of completeness we give the competitive ratios as a function of the prediction error. Purohit et al. [34], considered the *ski rental problem* and the *non-clairvoyant scheduling problem*. For both problems they gave algorithms that are both consistent and robust, and with a flavor similar to ours, the robustness and consistency of their algorithms are given as a function of some hyper parameter which has to be chosen by the algorithm in advance. For the ski rental problem in particular, Gollapudi et al. [17] considered the setting with multiple predictors, and designed and evaluated experimentally tight algorithms. Lykouris and Vassilvitskii [30] studied the *caching problem* (also known in the literature as *paging*), and were able to adapt the classical Marker algorithm [14] to obtain a trade-off between robustness and consistency, for this problem. Rohatgi [36] subsequently gave an algorithm whose competitive ratio has an improved dependence on the prediction errors. Further results in online algorithms with machine learned advice include the work by Lattanzi et al. [27] who studied the *restricted assignment scheduling problem* with predictions on some dual variables associated to the machines, and the work by Mitzenmacher [33] who considered a different *scheduling/queuing problem*. They introduced a novel quality measure for evaluating algorithms, called the *price of misprediction*. Finally, Mahdian et al. [31] studied problems where it is assumed that there exists an optimistic algorithm (which could in some way be interpreted as a prediction), and designed a meta-algorithm that interpolates between a worst-case algorithm and the optimistic one. They considered several problems, including the *allocation of online advertisement space*, and gave algorithms whose competitive ratio is also an interpolation between the competitive ratios of their corresponding optimistic and worst-case algorithms. However, the performance guarantee is not given as a function of the "prediction" error, but rather as a function of the respective ratios and the interpolation parameter.

## 2   Preliminaries

In this section we formally define the online algorithms of interestand define the so-called Lambert $W$-function that will be used in Section 3.

**Online algorithms with uniformly random arrivals.**   We briefly sketch some relevant definitions for the online problems that we consider in this work. We consider online selection problems in which the goal is to select the "best feasible" subset out of a finite set of objects $O$ with size $|O| = n$, that arrive online in a uniformly random order. More formally, the $n$ objects are revealed to the algorithm one object per round. In each round $i$, and upon revelation of the current object $o_i \in O$, the online selection algorithm has to irrevocably select an *outcome* $z_i$ out of a set of possible outcomes $Z(o_i)$ (which may depend on $o_1, o_2, \ldots o_i$ as well as $z_1, z_2, \ldots z_{i-1}$.) Each outcome $z_i$ is associated with a *value* $v_i(z_i)$, and all values $v_i$ become known to the algorithm with the arrival of $o_i$. The goal is to maximize the total value $T = \sum_i v_i(z_i)$.

The cost of an algorithm $\mathcal{A}$ selecting outcomes $z_1, z_2 \ldots z_n$ on $\sigma$ is defined as $T(\mathcal{A}(\sigma)) = \sum_i v_i(z_i)$. Such an algorithm $\mathcal{A}$ is $\gamma$-competitive if $\mathbb{E}(T(\mathcal{A}(\sigma))) \geq \gamma \cdot \text{OPT}(\sigma)$, for $0 < \gamma \leq 1$, where $\text{OPT}(\sigma)$ is the objective value of an offline optimal solution, i.e., one that is aware of the whole input sequence

$\sigma$ in advance. The expectation is taken over the randomness in $\sigma$ (and the internal randomness of $\mathcal{A}$ in case of a randomized algorithm). Alternatively, we say that $\mathcal{A}$ is a $\gamma$-approximation.

**The Lambert $W$-function.** The Lambert $W$-function is the inverse relation of the function $f(w) = we^w$. Here, we consider this function over the real numbers, i.e., the case $f : \mathbb{R} \to \mathbb{R}$. Consider the equation $ye^y = x$. For $-1/e \le x < 0$, this equation has two solutions denoted by $y = W_{-1}(x)$ and $y = W_0(x)$, where $W_{-1}(x) \le W_0(x)$ with equality if and only if $x = -1/e$.

Due to space constraints we only include a description of the algorithm for the classical secretary problem as well as a proof sketch for Theorem 1.2 in order to convey our ideas. Formal proofs, algorithm statements as well as more extensive discussions of the respective problems are deferred to the supplementary material.

## 3 Secretary problem

There are two versions of the secretary problem. In the *classical secretary problem*, the goal is to maximize the probability with which the best secretary is chosen. We consider a slightly different version, where the goal is to maximize the expected value of the chosen secretary. We refer to this as the *value-maximization secretary problem*.[6] In the remainder of this work, the term "secretary problem" will always refer to the value-maximization secretary problem, unless stated otherwise. The (optimal) solution [28, 11] to both variants of the secretary problem is to first observe a fraction of $n/e$ secretaries. After that, the first secretary with a value higher than the best value seen in that fraction is selected. This yields a $1/e$-approximation for both versions.

The machine learned advice that we consider in this section is a prediction $p^*$ for the maximum value $\text{OPT} = \max_i v_i$ among all secretaries. Note that we do not predict which secretary has the highest value. We define the *prediction error* as $\eta = |p^* - \text{OPT}|$. We emphasize that this parameter is not known to the algorithm, but is only used to analyse the algorithm's performance guarantee.

If we *a priori* know that the prediction error is small, then it is reasonable to pick the first element that has a value 'close enough' to the predicted optimal value. One small issue that arises here is that we do not know whether the predicted value is smaller or larger than the actual optimal value. In the latter case, we may "miss" the optimal element even if the prediction error was arbitrarily small. In order to circumvent this issue, one can first lower the predicted value $p^*$ slightly by some $\lambda > 0$ and then select the first element that is greater or equal than the threshold $p^* - \lambda$ (which is used to define the threshold parameter $t$ of in Phase II of Algorithm 1).[7] Roughly speaking, the parameter $\lambda$ can be interpreted as a guess for the prediction error $\eta$ or a confidence parameter in the prediction $p^*$. This allows us to interpolate between the following two extreme cases:
(i) If we have very low confidence in the prediction, we choose $\lambda$ close to $p^*$;
(ii) If we have high confidence in the prediction, we choose $\lambda$ close to 0.

In the first case, we essentially get back the classical solution [28, 11]. Otherwise, when the confidence in the prediction is high, we get a competitive ratio better than $1/e$ in case the prediction error $\eta$ is in fact small, in particular, smaller than $\lambda$. If our confidence in the prediction turned out to be wrong, when $\lambda$ is larger than the prediction error, we still obtain a $1/(ce)$-competitive algorithm. The parameter $c$ models what factor we are willing to lose in the worst-case when the prediction is poor (but the confidence in the prediction is high). We now give the pseudo-code of our algorithm and analyze it below in the proof sketch of Theorem 1.2.

Our algorithm can be seen a generalization of the well-known optimal algorithm for both the classical and value-maximization secretary problem [28, 11] that incorporates the idea sketched in the previous paragraph (while at the same time guaranteeing a constant competitive algorithm when the prediction error is large). Finally, if an $\exp\{W_0(-1/(ce))\}$-fraction of elements has been observed and nothing is selected yet, then Phase III is reached and we reset the threshold parameter $t$ ignoring the prediction. This serves as a fallback strategy to ensure a competitive ratio of $1/(ce)$, mimicking the optimal strategy for the value-maximization secretary problem.

**ALGORITHM 1:** Value-maximization secretary algorithm
---
**Input** : Prediction $p^*$ for (unknown) value $\max_i v_i$; confidence parameter $0 \leq \lambda \leq p^*$ and $c \geq 1$.
**Output :** Element $a$.

Set $v' = 0$.
**Phase I:**
**for** $i = 1, \ldots, \lfloor \exp\{W_{-1}(-1/(ce))\} \cdot n \rfloor$ **do**
⌊ Set $v' = \max\{v', v_i\}$
Set $t = \max\{v', p^* - \lambda\}$.
**Phase II:**
**for** $i = \lfloor \exp\{W_{-1}(-1/(ce))\} \cdot n \rfloor + 1, \ldots, \lfloor \exp\{W_0(-1/(ce))\} \cdot n \rfloor$ **do**
    **if** $v_i > t$ **then**
    ⌊ Select element $a_i$ and STOP.
Set $t = \max\{v_j \ : \ j \in \{1, \ldots, \lfloor \exp(W_0(-1/(ce))) \cdot n \rfloor\}\}$.
**Phase III:**
**for** $i = \lfloor \exp\{W_0(-1/(ce))\} \cdot n \rfloor + 1, \ldots, n$ **do**
    **if** $v_i > t$ **then**
    ⌊ Select element $a_i$ and STOP.
---

*Proof sketch of Theorem 1.2.* By carefully looking into the analysis of the classical secretary problem, see, e.g., [11, 28], it becomes clear that although initially observing an $1/e$-fraction of the items is the optimal trade-off for the classical algorithm and results in a competitive ratio of $1/e$, one could obtain a $1/(ce)$-competitive ratio (for $c > 1$) in two ways: by observing either less, or more items, more specifically an $\exp\{W_{-1}(-1/(ce))\}$ or $\exp\{W_0(-1/(ce))\}$ fraction of the items, respectively. These quantities arise as the two solutions of the equation $-x \ln x = 1/(ce)$.

We next discuss two lower bounds on the competitive ratio. First of all, we prove that in the worst-case we are always $1/(ce)$-competitive, while the second bound on the competitive ratio applies to cases in which the prediction error is small. In particular, suppose that $0 \leq \eta < \lambda$. We show that if $p^* > \text{OPT}$ and OPT appears in Phase II, then with probability of at least $f(c)$ we pick some element in Phase II of value at least $\text{OPT} - \lambda$. If on the other hand $p^* \leq \text{OPT}$, then with similar reasoning we show that with probability $f(c)$ we will pick some element in Phase II with value at least $\text{OPT} - \lambda - \eta$.

In any case, with probability at least $f(c)$, we will pick some element in Phase II with value at least $\min\{\text{OPT} - \lambda, \text{OPT} - \lambda - \eta\} = \text{OPT} - \lambda - \eta$ if $\eta < \lambda$. That is, if $0 \leq \eta < \lambda$, and if we assume that $\text{OPT} - \lambda - \eta \geq 0$, we are guaranteed to be $f(c)\left(1 - (\lambda + \eta)/\text{OPT}\right)$-competitive.                              □

# 4   Conclusion

Our results can be seen as the first evidence that online selection problems are a promising area for the incorporation of machine learned advice following the frameworks of [30, 34]. Many interesting problems and directions remain open. For example, does there exist a natural prediction model for the general matroid secretary problem? It is still open whether this problem admits a constant-competitive algorithm. Is it possible to show that there exists an algorithm under a natural prediction model that is constant-competitive for accurate predictions, and that is still $O(1/\log(\log(r)))$-competitive in the worst case, matching the results in [13, 26]? Furthermore, although our results are optimal within this specific three phased approach, it remains an open question whether they are optimal in general for the respective problems.

**BROADER IMPACT**

Matching problems play a very important role in crucial real-world situations, e.g., when assigning donor organs to patients in need, or assigning ventilators to hospitals during a lung virus pandemic. Although machine learning algorithms usually perform quite well on these type of problems, because of their critical importance, often a worst-case guarantee is paramount.

Our contribution follows a line of work that combines the power of machine learning with worst-case theoretical bounds to give a performance guarantee also for the rare instances on which machine learning algorithms are subpar.

## Acknowledgments and Disclosure of Funding

We would like to thank Kurt Mehlhorn for drawing our attention to the area of algorithms augmented with machine learned predictions. Part of this work was done while A. Antoniadis was at Saarland University and Max-Planck-Institute for Informatics, and was supported by DFG grant AN 1262/1-1.

## Footnotes

[1]This example is similar to the classical secretary problem [15].

[2] To be precise, we consider the so-called *value maximization* version of the problem, see Section 3.

[3] This corresponds to a prediction for the maximum price somebody is willing to offer in the laptop example.

[4] See the preliminaries section for a formal definition.

[5] The term "robustness" was used in a totally different sense in connection with the secretary problems [5].

[6] An $\alpha$-approximation for the classical secretary problem implies an $\alpha$-approximation for this problem.

[7] An interval around $p^*$ makes no difference, as long as only a prediction for the maximum value is given.

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
