[Supplementary Material]

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

We give detailed formulations of the meta-result in Theorem 3.1 for the secretary problem; in Theorem 4.1 for the online bipartite matching problem; and in Theorem 5.3 for the graphic matroid secretary problem.

In addition, we show that the online bipartite matching algorithm, given in Section 4, can be turned into a truthful mechanism in the case of the so-called *single-value unit-demand domains*. Details are given in Theorem F.2 in Appendix F. We show that the algorithm of Kesselheim et al. [22] can be turned into a truthful mechanism in the special case of *uniform edge weights*, where for every fixed online node in $L$, there is a common weight on all edges adjacent to it. In addition, we show that truthfulness can be preserved when predictions are included in the algorithm. We note that Reiffenhäuser [39] recently gave a truthful mechanism for the general online bipartite matching setting (without uniform edge weights). It would be interesting to see if her algorithm can be augmented with predictions as well.

**Remark 1.2.** *In the statements of Theorem 3.1, 4.1 and 5.3 it is assumed that the set of objects $O$ arriving online (either vertices or edges) is asymptotically large. We hide $o(1)$-terms at certain places for the sake of readability.*

Although the predictions provide very little information about the optimal solution, we are still able to obtain improved theoretical guarantees in the case where the predictions are sufficiently accurate. In the online bipartite matching setting with predictions for the nodes in $R$, we can essentially get close to a 1/2-approximation – which is best possible – assuming the predictions are close to perfect. This follows the fact that in this case we obtain the so-called *vertex-weighted online bipartite matching problem* for which there is a deterministic 1/2-competitive algorithm, and no algorithm can do better [1]. Roughly speaking, our algorithm converges to that in [1] when the predictions get close

to perfect. This will be discussed further in Section 4. For the graphic matroid secretary problem, we are also able to get close to a 1/2-approximation in the case where the predictions (the maximum edge weights adjacent to the nodes in the graph) get close to perfect. We note that this is probably not tight. We suspect that, when given perfect predictions, it is possible to obtain an algorithm with a better approximation guarantee. This is an interesting open problem.

## 1.2 Related work

This subsection consists of three parts. First we discuss relevant approximation algorithms for the matroid secretary problem without any form of prior information, then we consider models that incorporate additional information, such as the area of prophet inequalities. Finally, we give a short overview of related problems that have been analyzed with the inclusion of machine learned advice following the frameworks in [32, 38], which we study here as well.

**Approximation algorithms for the matroid secretary problem.** The classical secretary problem was originally introduced by Gardner [15], and solved by Lindley [30] and Dynkin [11], who gave $1/e$-competitive algorithms. Babaioff et al. [4] introduced the matroid secretary problem, a considerable generalization of the classical secretary problem, where the goal is to select a set of secretaries with maximum total value under a matroid constraint for the set of feasible secretaries. They provided an $O(1/\log(r))$-competitive algorithm for this problem, where $r$ is the rank of the underlying matroid. Lachish [28] later gave an $O(1/\log\log(r))$-competitive algorithm, and a simplified algorithm with the same guarantee was given by Feldman, Svensson and Zenklusen [13]. It is still a major open problem if there exists a constant-competitive algorithm for the matroid secretary problem. Nevertheless, many constant-competitive algorithms are known for special classes of matroids, and we mention those relevant to the results in this work (see, e.g., [4, 41] for further related work).

Babaioff et al. [4] provided a 1/16-competitive algorithm for the case of transversal matroids, which was later improved to a 1/8-competitive algorithm by Dimitrov and Plaxton [9]. Korula and Pál [25] provided the first constant competitive algorithm for the online bipartite matching problem considered in Section 4, of which the transversal matroid secretary problem is a special case. In particular, they gave a 1/8-approximation. Kesselheim et al. [22] provided a $1/e$-competitive algorithm, which is best possible, as discussed above.

For the graphic matroid secretary problem, Babaioff et al. [4] provide a deterministic 1/16-competitive algorithm. This was improved to a $1/(3e)$-competitive algorithm by Babaioff et al. [3]; a $1/(2e)$-competitive algorithm by Korula and Pál [25]; and a 1/4-competitive algorithm by Soto et al. [41], which is currently the best algorithm. The algorithm from [4] is deterministic, whereas the other three are randomized.

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

## 2.2 Graph theoretical notation

An undirected graph $G = (V, E)$ is defined by a set of nodes $V$ and set of edges $E \subseteq \{\{u, v\} : u, v \in V, u \neq v\}$. A bipartite graph $G = (L \cup R, E)$ is given by two sets of nodes $L$ and $R$, and

$E \subseteq \{\{\ell, r\} : \ell \in L, r \in R\}$. In the bipartite case we sometimes write $(\ell, r)$ to indicate that $\ell \in L$ and $r \in R$ (we also use this notation for directed arcs in general directed graphs). For a set of nodes $W$, we use $G[W]$ to denote the induced (bipartite) subgraph on the nodes in $W$.

A function $w : E \to \mathbb{R}_{\geq 0}$ is called a weight function on the edges in $E$; we sometimes write $w(\ell, r)$ in order to denote $w(\{u, v\})$ for $\{u, v\} \in E$. A matching $M \subseteq E$ is a subset of edges so that every node is adjacent to at most one edge in $M$. For a set of nodes $W$, we write $W[M]$ to denote the nodes in $W$ that are adjacent to an edge in $M$. Such nodes are said to be matched. If $G$ is undirected, we say that $M$ is perfect if every node in $V$ is adjacent to precisely one edge in $M$. If $G$ is bipartite, we say that $M$ is perfect w.r.t $L$ if every $\ell \in L$ is adjacent to one edge in $M$, and perfect w.r.t $R$ if every $r \in R$ is adjacent to some edge in $M$.

**Remark 2.1.** *When $G$ is bipartite, we will assume that for every subset $S \subseteq L$, there is a perfect matching w.r.t $S$ in $G[S \cup R]$. This can be done without loss of generality by adding for every $\ell \in L$ a node $r^\ell$ to the set $R$, and adding the edge $\{\ell, r^\ell\}$ to $E$. Moreover, given a weight function on $E$ we extend it to a weight function on $E'$ by giving all the new edges weight zero.*

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

**Remark 3.2.** *Note, that the bound obtained in Theorem 3.1 has a discontinuity at $\eta = \lambda$. This can be easily smoothed out by selecting $\lambda$ according to some distribution, which now represents our confidence in the prediction $p^*$. The competitive ratio will start to drop earlier in this case, and will continuously reach $1/(ce)$. Furthermore, for $\eta = \lambda = 0$ this bound is tight for any fixed c. We illustrate how the competitive ratio changes as a function of $\eta$ in Appendix A.1.*

*Moreover, in Appendix A.2, we show that our deterministic Algorithm 1 can achieve a better competitive ratio than its corresponding naive randomization.*

*Proof of Theorem 3.1.* By carefully looking into the analysis of the classical secretary problem, see, e.g., [11, 30], it becomes clear that although sampling an $1/e$-fraction of the items is the optimal trade-off for the classical algorithm and results in a competitive ratio of $1/e$, one could obtain a $1/(ce)$-competitive ratio (for $c > 1$) in two ways: by sampling either less, or more items, more specifically an $\exp\{W_{-1}(-1/(ce))\}$ or $\exp\{W_0(-1/(ce))\}$ fraction of the items respectively. These quantities arise as the two solutions of the equation

$$-x \ln x = \frac{1}{ce}.$$

We next provide two lower bounds on the competitive ratio.

First of all, we prove that in the worst-case we are always $1/(ce)$-competitive. We consider two cases.

*Case 1: $p^* - \lambda > OPT$.* Then we never pick an element in Phase II, which means that the algorithm is equivalent to the algorithm that observes a fraction $\exp\{W_0(-1/(ce))\}$ of all elements and then chooses the first element better than what has been seen before, which we know is $1/(ce)$-competitive.

*Case 2. $p^* - \lambda \le OPT$.* Consider a fixed arrival order and suppose that, for this permutation, we select OPT in the algorithm that first observes a fraction $\exp\{W_{-1}(-1/(ce))\}$ of all elements and then selects the first element better than what has been seen before (which we know is $1/(ce)$-competitive). It should be clear that our algorithm also chooses OPT in this case. As the analysis in [11, 30] relies on analyzing the probability with which we pick OPT, it follows that our algorithm is also $1/(ce)$-competitive in this case.

The second bound on the competitive ratio applies to cases in which the prediction error is small. In particular, suppose that $0 \le \eta < \lambda$.

*Case 1: $p^* > OPT$.* We know that $p^* - \lambda < OPT$, as $\eta < \lambda$. Therefore, if $OPT$ appears in Phase II, and we have not picked anything so far, we will pick $OPT$. Since $OPT$ appears in Phase II with probability $f(c)$, we in particular pick some element in Phase II with value at least $OPT - \lambda$ with probability $f(c)$ (note that this element does not have to be $OPT$ necessarily).

*Case 2: $p^* \le OPT$.* In this case, using similar reasoning as in Case 1, with probability $f(c)$ we will pick some element with value at least $OPT - \lambda - \eta$. To see this, note that in the worst case we would have $p^* = OPT - \eta$, and we could select an element with value $p^* - \lambda$, which means that the value of the selected item is $OPT - \lambda - \eta$.

This means that, in any case, with probability at least $f(c)$, we will pick some element in Phase II with value at least $\min\{OPT - \lambda, OPT - \lambda - \eta\} = OPT - \lambda - \eta$ if $\eta < \lambda$. That is, if $0 \le \eta < \lambda$, and if we assume that $OPT - \lambda - \eta \ge 0$, we are guaranteed to be $f(c) (1 - (\lambda + \eta)/OPT)$-competitive. $\square$

## 4 Online bipartite matching with random arrivals

In this section we consider a generalization of the value-maximization secretary problem discussed in Section 3. We study an online bipartite matching problem on a graph $G = (L \cup R, E)$ with edge weight function $w : E \to \mathbb{R}_{\ge 0}$. The vertices $\ell \in L$ arrive in a uniformly random order. Whenever a vertex $\ell \in L$ arrives, it reveals its neighbors $r \in R$ and what the corresponding edge weights $w(\ell, r)$ are. We

then have the option to add an edge of the form $(\ell, r)$, provided $r$ has not been matched in an earlier step. The goal is to select a set of edges, i.e., a matching, with maximum weight.

We assume that we are given, for all offline nodes $r \in R$, a prediction $p_r^*$ for the value of the edge weight adjacent to $r$ in some fixed optimal offline matching (which is zero if $r$ is predicted not to be matched in this offline matching). That is, we predict that there exists some fixed optimal offline matching in which $r$ is adjacent to an edge of weight $p_r^*$ without predicting which particular edge this is. Note that the predictions $p = (p_1^*, \ldots, p_r^*)$ implicitly provide a prediction for OPT, namely $\sum_{r \in R} p_r^*$.

It turns out that this type of predictions closely corresponds to the so-called *online vertex-weighted bipartite matching problem* where every offline node is given a weight $w_r$, and the goal is to select a matching with maximum weight, which is the sum of all weights $w_r$ for which the corresponding $r$ is matched in the online algorithm. This problem has both been studied under adversarial arrivals [20, 1] and uniformly random arrivals [34, 18]. In case the predictions are perfect, then, in order to find a matching with the corresponding predicted values, we just ignore all edges $w(\ell, r)$ that do not match the value $p_r^*$. This brings us in a special case of the online vertex-weighted bipartite matching problem.

The prediction error in this section will be defined as the maximum error over all predicted values and the minimum over all optimal matchings in $G$. We use $\mathcal{M}(G, w)$ to denote the set of all optimal matchings in $G$ with respect to the weight function $w$, and then define

$$\eta = \min_{M \in \mathcal{M}(G, r)} \max_{r \in R} |p_r^* - w(M_r)|.$$

Here, we use $w(M_r)$ to denote the value of the edge adjacent to $r \in R$ in a given optimal solution with objective value $\text{OPT} = \sum_r w(M_r)$.

In the next sections we will present deterministic and randomized algorithms, inspired by algorithms for the online vertex-weighted bipartite matching problem, that can be combined with the algorithm in [22] in order to obtain algorithms that incorporate the predictions and have the desired properties. We start with a deterministic algorithm, which is the main result of this section.

## 4.1 Deterministic algorithm

We first give a simple deterministic greedy algorithm that provides a 1/2-approximation in the case when the predictions are perfect (which is true even for an adversarial arrival order of the nodes). It is very similar to a greedy algorithm given by Aggarwal et al. [1]. Although we do not emphasize it in the description, this algorithm can be run in an online fashion.

---
**ALGORITHM 2:** Threshold greedy algorithm
---
**Input** : Thresholds $t = (t_1, \ldots, t_{|R|})$ for offline nodes $r \in R$; ordered list $(v_1, \ldots, v_\ell) \subseteq L$.
**Output:** Matching $M$

Set $M = \emptyset$.
**for** $i = 1, \ldots, \ell$ **do**
    Set $r^i = \text{argmax}_r \{w(v_i, r) : r \in \mathcal{N}(v_i), w(v_i, r) \geq t_r \text{ and } r \notin R[M]\}$.
    **if** $r^i \neq \emptyset$ **then**
    $\quad |$ Set $M = M \cup \{v_i, r^i\}$.
    **end**
**end**

---

Provided that there exists an offline matching in which every $r \in R$ is adjacent to some edge with weight at least $t_r$, it can be shown, using the same arguments as given in [1], that the threshold

greedy algorithm yields a matching with weight at least $\frac{1}{2}\sum_r t_r$. We present the details in the proof of Theorem 4.1 later on.

It is also well-known that, even for uniformly random arrival order and unit edge weights, one cannot obtain anything better than a 1/2-approximation with a deterministic algorithm [1].[8] This also means that, with our choice of predictions, we cannot do better than a 1/2-approximation in the ideal case in which the predictions are perfect. Therefore, our aim is to give an algorithm that includes the predictions in such a way that, if the predictions are good (and we have high confidence in them), we should approach a 1/2-approximation, whereas if the predictions turn out to be poor, we are allowed to lose at most a constant factor w.r.t. the $1/e$-approximation in [22].

Algorithm 3 is a deterministic algorithm satisfying these properties, that, similar to Algorithm 1, consists of three phases. The first two phases correspond to the algorithm of Kesselheim et al. [22]. In the third phase, we then run the threshold greedy algorithm as described in Algorithm 2. Roughly speaking, we need to keep two things in mind. First of all, we should not match up too many offline nodes in the second phase, as this would block the possibility of selecting a good solution in the third phase in case the predictions are good. On the other hand, we also do not want to match too few offline nodes in the second phase, otherwise we are no longer guaranteed to be constant-competitive in case the predictions turn out to be poor. The analysis of Algorithm 3 given in Theorem 4.1 shows that it is possible to achieve both these properties.

For the sake of simplicity, in both the description of Algorithm 3 and its analysis in Theorem 4.1, we use a common $\lambda$ to lower the predicted values (as we did in Section 3 for the secretary problem). Alternatively, one could use a resource-specific value $\lambda_r$ for this as well.

---

**ALGORITHM 3:** Online bipartite matching algorithm with predictions

**Input** : Predictions $p^* = (p_1^*, \ldots, p_{|R|}^*)$, confidence parameter $0 \le \lambda \le \min_r p_r^*$, and $c > d \ge 1$.
**Output:** Matching $M$.

**Phase I:**                                                                        /*Algorithm from [22]
**for** $i = 1, \ldots, \lfloor n/c \rfloor$ **do**
| Observe arrival of node $\ell_i$, and store all the edges adjacent to it.
**end**
Let $L' = \{\ell_1, \ldots, \ell_{\lfloor n/c \rfloor}\}$ and $M = \emptyset$.
**Phase II:**
**for** $i = \lfloor n/c \rfloor + 1, \ldots, \lfloor n/d \rfloor$ **do**
| Set $L' = L' \cup \ell_i$.
| Set $M^i$ = optimal matching on $G[L' \cup R]$.
| Let $e^i = (\ell_i, r)$ be the edge assigned to $\ell_i$ in $M^i$.
| **if** $M \cup e^i$ *is a matching* **then**
| | Set $M = M \cup \{e^i\}$.
| **end**
**end**
**Phase III:**                                                                      /*Threshold greedy algorithm
**for** $i = \lfloor n/d \rfloor + 1, \ldots, n$ **do**
| Set $r^i = \operatorname{argmax}_r \{w(v_i, r) : r \in \mathcal{N}(v_i), w(v_i, r) \ge p_r^* - \lambda \text{ and } r \notin R[M]\}$
| **if** $r^i \ne \emptyset$ **then**
| | Set $M = M \cup \{\ell_i, r\}$.
| **end**
**end**

---

**Theorem 4.1.** *For any $\lambda \ge 0$ and $c > d \ge 1$, there is a deterministic algorithm for the online bipartite matching problem with uniformly random arrivals that is asymptotically $g_{c,d,\lambda}(\eta)$-competitive*

*in expectation, where*

$$g_{c,d,\lambda}(\eta) = \left\{ \begin{array}{ll} \max\left\{ \frac{1}{c}\ln(\frac{c}{d}), \left[\frac{d-1}{2c}\left(\max\left\{1 - \frac{(\lambda+\eta)|\psi|}{OPT}, 0\right\}\right)\right]\right\} & \text{if } 0 \le \eta < \lambda, \\ \frac{1}{c}\ln(\frac{c}{d}) & \text{if } \eta \ge \lambda. \end{array}\right\},$$

*and $|\psi|$ is the cardinality of an optimal (offline) matching $\psi$ of the instance.*

If $\lambda$ is small, we roughly obtain a bound of $(d-1)/2c$ in case $\eta$ is small as well, and a bound of $\ln(c/d)/c$ if $\eta$ is large. Moreover, when $c/d \to 1$, we approach a bound of $1/2$ in case $\eta$ is small, whereas the worst-case guarantee $\ln(c/d)/c$ for large $\eta$ then increases.

*Proof of Theorem 4.1.* We provide two lower bounds on the expected value of the matching $M$ output by Algorithm 3.

First of all, the analysis of the algorithm of Kesselheim et al. [22] can be generalized to the setting we consider in the first and second phase. In particular, their algorithm then yields a

$$\left(\frac{1}{c} - \frac{1}{n}\right) \ln\left(\frac{c}{d}\right)\text{-competitive approximation.}$$

For completeness, we present a proof of this statement in Appendix C.

The second bound we prove on the expected value of the matching $M$ is based on the threshold greedy algorithm we use in the third phase. Let $\psi \in \mathcal{M}(G, w)$ be an optimal (offline) matching, with objective value OPT, and suppose that

$$\eta = \max_{r \in R} |\psi_r - p_r^*| < \lambda.$$

The proof of the algorithm in [22] analyzes the expected value of the online vertices in $L$. Here we take a different approach and study the expected value of the edge weights adjacent to the nodes in $r \in R$. Fix some $r \in R$ and consider the edge $(\ell, r)$ that is matched to $r$ in the optimal offline matching $\psi$ (if any).

Let $X_r$ be a random variable denoting the value of node $r \in R$ in the online matching $M$ chosen by Algorithm 3. Let $Y_\ell$ be a random variable that denotes the value of node $\ell \in L[\psi]$ in the online matching $M$. It is not hard to see that

$$\mathbb{E}(M) = \sum_{r \in R} \mathbb{E}(X_r) \quad \text{and} \quad \mathbb{E}(M) \ge \sum_{\ell \in L[\psi]} \mathbb{E}(Y_\ell). \tag{1}$$

For the inequality, note that for any fixed permutation the value of the obtained matching is always larger or equal to the sum of the values that were matched to the nodes $\ell \in L[\psi]$.

Now, consider a fixed $\ell$ and $r$ for which the edge $(\ell, r)$ is contained in $\psi$. We will lower bound the expectation $\mathbb{E}(X_r + Y_\ell)$ based on the expected value these nodes would receive, roughly speaking, if they get matched in the third phase. Therefore, suppose for now that $r$ did not get matched in the second phase, and that $\ell$ appears in the part of the uniformly random permutation considered in the third phase. We will later lower bound the probability with which the events occur. By definition of the greedy threshold algorithm, we know that at the end of Phase III either node $r \in R$ is matched, or otherwise at least node $\ell$ is matched to some other $r' \in R$ for which

$$w(\ell, r') \ge w(\ell, r) \ge p_r^* - \eta \ge p_r^* - \lambda.$$

To see this, note that when the vertex $\ell$ arrived, there was the option to match it to $r$, as this node is assumed not to have been matched in the second phase. So, there are the following three cases: either

$\ell$ got matched to $r$, or it got matched to some other $r'$ for which $w(\ell, r') \geq w(\ell, r) \geq p_r^* - \lambda$ or $r$ was matched earlier during the third phase to some other $\ell'$ for which $w(\ell', r) \geq p_r^* - \lambda$.

Looking closely at the analysis of Kesselheim et al. [22], see Appendix C.3, it follows that the probability that a fixed node $r$ did not get matched in the second phase satisfies

$$\text{P}(r \text{ was not matched in Phase II}) \geq \frac{d}{c} - o(1).$$

This lower bound is true independent of whether or not $\ell$ appeared in Phase III, or the first two phases (see Appendix C). This former event holds with probability $(1 - 1/d)$. Therefore,

$$\text{P}(r \text{ was not matched in Phase II } and \, \ell \text{ arrives in Phase III}) \geq \left(\frac{d}{c} - o(1)\right)\left(1 - \frac{1}{d}\right) = \frac{d-1}{c} - o(1).$$

Furthermore, we have from an earlier argument that under this condition either $X_r \geq p_r^* - \lambda$ or $Y_\ell \geq p_r^* - \lambda$. This implies that

$$\mathbb{E}(X_r + Y_\ell) \geq \left(\frac{d-1}{c} - o(1)\right)(p_r^* - \lambda), \tag{2}$$

and combining this with (1), we find

$$2 \cdot \mathbb{E}(M) \geq \left(\frac{d-1}{c} - o(1)\right) \sum_{r \in R[\psi]} (p_r^* - \lambda) \geq \left(\frac{d-1}{c} - o(1)\right) (\text{OPT} - (\lambda + \eta)|\psi|)$$

assuming that $\text{OPT} - (\lambda + \eta)|\psi| \geq 0$. Rewriting this gives

$$\mathbb{E}(M) \geq \left(\frac{d-1}{2c} - o(1)\right)\left(1 - \frac{(\lambda + \eta)|\psi|}{\text{OPT}}\right) \cdot \text{OPT},$$

which yields the desired bound. □

In order to get close to a $1/2$-approximation in case the predictions are (almost) perfect, we have to choose both $c$ and $d$ very large, as well as the ratio $c/d$ close to 1 (but still constant). It is perhaps also interesting to note that Theorem 4.1 does not seem to hold in case we interchange the second and third phase. In particular, if the predictions are too low, we most likely match up too many nodes in $r \in R$ already in the second phase (that now would execute the threshold greedy algorithm).

## 4.2 Truthful mechanism for single-value unit-demand domains

Algorithm 3 can be turned into a truthful mechanism for *single-value unit-demand domains*. Here, every node arriving online corresponds to an *agent* $i$ that is interested in a subset $A_i$ of the *items* in $R$ (i.e., her neighbors in $R$) between which she is indifferent. That is, agent $i$ has a value $v_i$ for all items in $A_i$, and zero for all other items. Whenever an agent arrives, she reports a value $v_i'$ for the items in $A_i$ that she is interested in (we assume that these sets are common knowledge). Based on the value $v_i'$, the mechanism decides if it wants to allocate one of the items in $A_i$ (not yet assigned to any agent) to $i$, and, if so, sets a price $\rho_i$ for this item that is charged to agent $i$. The goal is to choose an allocation that maximizes the social welfare (which is the total value of all assigned items).

The goal of the agents is to maximize their utility which is their valuation minus the price that they are charged by the mechanism for the item assigned to them (if any). We want to design a mechanism that incentivizes agents to report their true value $v_i$.

In Appendix F, we provide formal definitions of all notions and show that Algorithm 3 can be turned into a truthful mechanism for which its social welfare guarantee is $g_{\lambda,c,d}(\eta)$ as in Theorem 4.1. In particular, we do this by exploiting some of the freedom we have in Algorithm 3.

## 4.3 Randomized algorithm

If we allow randomization, we can give better approximation guarantees than the algorithm given in the previous section by using a convex combination of the algorithm of Kesselheim et al. [22], and the randomized algorithm of Huang et al. [18] for online vertex-weighted bipartite matching with uniformly random arrivals.

We give a simple, generic way to reduce an instance of online bipartite matching with predictions $p_r^*$ for $r \in R$ to an instance of online vertex-weighted bipartite matching with vertex weights (that applies in case the predictions are accurate). This reduction works under both a uniformly random arrival order of the vertices, as well as an adversarial arrival order.

Suppose we are given an algorithm $\mathcal{A}$ for instances of the online vertex-weighted bipartite matching problem. Now, for the setting with predictions, fix some parameter $\lambda > 0$ up front. Whenever a vertex $\ell$ arrives online we only take into account edges $(\ell, r)$ with the property that $w(\ell, r) \in [p_r^* - \lambda, p_r^* + \lambda]$, and ignore all edges that do not satisfy this property. We then make the same decision for $\ell$ as we would do in algorithm $\mathcal{A}$ (only considering the edges that satisfy the interval constraint given above) based on assuming those edges have weight $p_r^*$. Then the matching that algorithm $\mathcal{A}$ generates has the property that its objective value is close (in terms of $\lambda$ and $\eta$) to the objective value under the original weights (and this remains to hold true in expectation in case of a uniformly random arrival order of the vertices).

The 0.6534-competitive algorithm of Huang et al. [18] is the currently best known randomized algorithm $\mathcal{A}$, for online vertex-weighted bipartite matching with uniformly random arrivals, and can be used for our purposes. Detailed statements will be given in the journal version of this work.

## 5 Deterministic graphic matroid secretary algorithm

In this section we will study the graphic matroid secretary problem. Here, we are given a (connected) graph $G = (V, E)$ of which the edges in $E$ arrive online. There is an edge weight function $w : 2^E \to \mathbb{R}_{\geq 0}$ and a weight is revealed if an edge arrives. The goal is to select a forest (i.e., a subset of edges that does not give rise to a cycle) of maximum weight. The possible forests of $G$ form the *independent sets of the graphical matroid on $G$*. It is well-known that the offline optimal solution of this problem can be found by the greedy algorithm that orders all the edge weights in decreasing order, and selects elements in this order whenever possible.

We will next explain the predictions that we consider in this section. For every node $v \in V$, we let $p_v^*$ be a prediction for the *maximum edge weight* $\max_{u \in \mathcal{N}(v)} w_{uv}$ adjacent to $v \in V$. The prediction error is defined by

$$\eta = \max_{v \in V} \left| p_v^* - w_{\max}(v) \right|, \quad \text{where} \quad w_{\max}(v) = \max_{u \in \mathcal{N}(v)} w_{uv}.$$

**Remark 5.1.** *Although the given prediction is formulated independently of any optimal solution (as we did in the previous sections), it is nevertheless equivalent to a prediction regarding the maximum weight $w_{uv}$ adjacent to $v \in V$ in an optimal (greedy) solution. To see this, note that the first time the offline greedy algorithm encounters an edge weight $w_{uv}$ adjacent to $v$, it can always be added as currently there is no edge adjacent to $v$. So adding the edge $\{u, v\}$ cannot create a cycle.*

Before we give the main result of this section, we first provide a *deterministic $(1/4 - o(1))$-competitive* algorithm for the graphic matroid secretary problem in Section 5.1, which is of independent interest. Asymptotically, this is an improvement over the algorithm of Soto et al. [41] in the sense that we do not require randomness in our algorithm. We then continue with an algorithm incorporating the predictions in Section 5.2.

## 5.1 Deterministic approximation algorithm

In this section, we provide a deterministic $(1/4-o(1))$-approximation for the graphic matroid secretary problem. For a given undirected graph $G = (V, E)$, we use the bipartite graph interpretation that was also used in [4]. That is, we consider the bipartite graph $B_G = (E \cup V, A)$, where an edge $\{e, v\} \in A$, for $e \in E$ and $v \in V$, if and only if $v \in e$. Note that this means that every $e = \{u, v\}$ is adjacent to precisely $u$ and $v$ in the bipartite graph $B_G$. Moreover, the edge weights for $\{e, v\}$ and $\{e, u\}$ are both $w_e$ (which is revealed upon arrival of the element $e$).[9] We emphasize that in this setting, the $e \in E$ are the elements that arrive online.

Algorithm 4 is very similar to the algorithm in [22] with the only difference that we allow an edge $\{e, u\}$ or $\{e, v\}$ to be added to the currently selected matching $M$ in $B_G$ if and only if *both* nodes $u$ and $v$ are currently not matched in $M$. In this section we often represent a (partial) matching in $B_G$ by a directed graph (of which its undirected counterpart does not contain any cycle). In particular, given some matching $M$ in $B_G$, we consider the directed graph $D_M$ with node set $V$. There is a directed edge $(u, v)$ if and only if $\{e, v\}$ is an edge in $M$, where $e = \{u, v\} \in E$. Note that every node in $D_M$ has an in-degree of at most one as $M$ is a matching.

Using the graph $D_M$ it is not too hard to see that if both $u$ and $v$ are not matched in the current matching $M$, then adding the edge $\{e, u\}$ or $\{e, v\}$ can never create a cycle in the graph formed by the elements $e \in E$ matched up by $M$, called $E[M]$, which is the currently chosen independent set in the graphic matroid. This subgraph of $G$ is precisely the undirected counterpart of the edges in $D_M$ together with $\{u, v\}$. For sake of contradiction, suppose adding the $\{u, v\}$ to $E[M]$ would create an (undirected) cycle $C$. As both $u$ and $v$ have in-degree zero (as they are unmatched in $M$), it follows that some node on the cycle $C$ must have two incoming directed edges in the graph $D_M$. This yields a contradiction.

We note that, although $u$ and $v$ being unmatched is sufficient to guarantee that the edge $\{u, v\}$ does not create a cycle, this is by no means a necessary condition.

---

**ALGORITHM 4:** Deterministic graphic matroid secretary algorithm

---

**Input** : Bipartite graph $G_B = (E \cup V, A)$ for undirected weighted graph $G = (V, E)$ with $|E| = m$.
**Output:** Matching $M$ of $G_B$ corresponding to forest in $G$.

**Phase I:**
**for** $i = 1, \ldots, \lfloor m/c \rfloor$ **do**
|    Observe arrival of element $e_i$, but do nothing.
**end**
Let $E' = \{e_1, \ldots, e_{\lfloor m/c \rfloor}\}$ and $M = \emptyset$.
**Phase II:**
**for** $i = \lfloor m/c \rfloor + 1, \ldots, m$ **do**
|    Let $E' = E' \cup e_i$.
|    Let $M^i =$ optimal matching on $B_G[E' \cup V]$.
|    Let $a^i = \{e_i, u\}$ be the edge assigned to $e_i = \{u, v\}$ in $M^i$ (if any).
|    **if** $M \cup a^i$ *is a matching and both* $u$ *and* $v$ *are unmatched in* $M$ **then**
|      | Set $M = M \cup a^i$.
|    **end**
**end**

---

**Theorem 5.2.** *Algorithm 4 is a deterministic $(1/4 - o(1))$-competitive algorithm for the graphic matroid secretary problem.*

**ALGORITHM 5:** Graphic matroid secretary algorithm with predictions

---

**Input** : Bipartite graph $G_B = (E \cup V, A)$ for undirected graph $G = (V, E)$ with $|E| = m$. Predictions $p = (p_1^*, \ldots, p_n^*)$. Confidence parameter $0 \leq \lambda \leq \min_i p_i^*$ and $c > d \geq 1$.

**Output:** Matching $M$ of $G_B$ corresponding to forest in $G$.

**Phase I:**
**for** $i = 1, \ldots, \lfloor m/c \rfloor$ **do**
    Let $e_i = \{u, v\}$.
    Set $t_v = \max\{t_v, w(u,v)\}$ and $t_u = \max\{t_u, w(u,v)\}$.
**end**
Let $E' = \{e_1, \ldots, e_{\lfloor m/c \rfloor}\}$ and $M = \emptyset$.

**Phase II:**
**for** $i = \lfloor m/c \rfloor + 1, \ldots, \lfloor m/d \rfloor$ **do**
    Let $e_i = \{u, v\}$, $S = \{x \in \{u, v\} : x \notin E[M] \text{ and } w(u,v) \geq \max\{t_x, p_x^* - \lambda\}\}$ and $y_i = \text{argmax}_{x \in S} p_x^* - \lambda$.
    **if** $E[M] \cup \{e_i\}$ *does not contain a cycle* **then**
        Set $M = M \cup \{e_i, y_i\}$.
    **end**
**end**

**Phase III:**
**for** $i = \lfloor m/d \rfloor + 1, \ldots, m$ **do**
    Let $E' = E' \cup e_i$.
    Let $M^i$ = optimal matching on $B_G[E' \cup V]$.
    Let $a^i = \{e_i, u\}$ be the edge assigned to $e_i = \{u, v\}$ in $M^i$ (if any).
    **if** $M \cup a^i$ *is a matching and both $u$ and $v$ are unmatched in $M$* **then**
        Set $M = M \cup a^i$.
    **end**
**end**

---

## 5.2 Algorithm including predictions

In this section we will augment Algorithm 4 with the predictions for the maximum edge weights adjacent to the nodes in $V$. We will use the bipartite graph representation $B_G$ as introduced in Section 5. Algorithm 5 consists of three phases, similar to Algorithm 3.

Instead of exploiting the predictions in Phase III, we already exploit them in Phase II for technical reasons.[10] Roughly speaking, in Phase II, we run a greedy-like algorithm that selects for every node $v \in V$ at most one edge that satisfies a threshold based on the prediction for node $v$. In order to guarantee that we do not select too many edges when the predictions are poor (in particular when they are too low), we also include a ''fail-safe' threshold based on the edges seen in Phase I.

**Theorem 5.3.** *For any $\lambda \geq 0$ and $c > d \geq 1$, there is a deterministic algorithm for the graphic matroid secretary problem that is asymptotically $g_{c,d,\lambda}(\eta)$-competitive in expectation, where*

$$
g_{c,d,\lambda}(\eta) = \begin{cases} \max\left\{\frac{d-1}{c^2}, \frac{1}{2}\left(\frac{1}{d} - \frac{1}{c}\right)\left(1 - \frac{2(\lambda+\eta)|V|}{OPT}\right)\right\} & \text{if } 0 \leq \eta < \lambda, \\ \frac{d-1}{c^2} & \text{if } \eta \geq \lambda. \end{cases}
$$

If $\lambda$ is small, we roughly obtain a bound of $(1/d - 1/c)/2$ in case $\eta$ is small, and a bound of $(d-1)/c^2$ if $\eta$ is large. Note that the roles of $c$ and $d$ have interchanged w.r.t. Algorithm 3 as we now exploit the predictions in Phase II instead of Phase III. Roughly speaking, if $d \to 1$ and $c \to \infty$ we approach a bound of $1/2$ if the predictions are good, whereas the bound of $(d-1)/c^2$ becomes arbitrarily bad.

*Proof of Theorem 5.3.* As in the proof of Theorem 4.1, we provide two lower bounds on the expected value on the matching $M$ outputted by Algorithm 5. We first provide a bound of

$$\frac{1}{2}\left(\frac{1}{d} - \frac{1}{c}\right)\left(1 - \frac{2(\lambda + \eta)|V|}{\text{OPT}}\right)\text{OPT}$$

in case the prediction error is small, i.e., when $\eta < \lambda$. For simplicity we focus on the case where for each $v \in V$, the weights of the edges adjacent to $v$ in $G$ are distinct.[11]

For every $v \in V$, let $e_{max}(v)$ be the (unique) edge adjacent to $v$ with maximum weight among all edges adjacent to $v$. Consider the fixed order $(e_1, \ldots, e_m)$ in which the elements in $E$ arrive online, and define $Q = \{v \in V : e_{max}(v) \text{ arrives in Phase II}\}$. We will show that the total weight of all edges selected in Phase II is at least $\frac{1}{2}\sum_{v \in Q}(p_v^* - (\lambda + \eta))$. Let $T \subseteq Q$ be the set of nodes for which the edge $e_{max}(v)$ arrives in Phase II, but for which $v$ does not get matched up in Phase II.

In particular, let $v \in T$ and consider the step $\ell$ in Phase II in which $e_{max}(v) = \{u, v\}$ arrived. By definition of $\eta$, and because $\eta < \lambda$, we have

$$w(u, v) = w_{\max}(v) \geq \max\{t_v, p_v^* - \eta\} \geq \max\{t_v, p_v^* - \lambda\}, \tag{3}$$

and so the pair $\{e_i, v\}$ is eligible (in the sense that $v \in S$). Since $v$ did not get matched, one of the following two holds:

i) The edge $e_{\max}(v)$ got matched up with $u$.

ii) Adding the edge $\{e_{\max}(v), v\}$ to $M$ would have yielded a cycle in $E[M] \cup e_{\max}(v)$.

Note that it can never be the case that we do *not* match up $e_{\max}(v)$ to $u$ for the reason that it would create a cycle. This is impossible as both $u$ and $v$ are unmatched.

Now, in the first case, since $u$ is matched it must hold that $w(u, v) \geq \max\{t_u, p_u^* - \lambda\}$, and $p_u^* - \lambda \geq p_v^* - \lambda$ as $v$ was eligible to be matched up in the online matching $M$ (but it did not happen). Further, combining (3) and the definition of $y_i$ in Phase II, yields

$$2w(u, v) \geq (p_u^* - \lambda) + (p_v^* - \eta) \geq (p_u^* - \eta - \lambda) + (p_v^* - \eta - \lambda). \tag{4}$$

We call $u$ the *i)-proxy* of $v$ in this case.

In the second case, if adding $e_{\max}(v)$ would have created an (undirected) cycle in the set of elements (i.e., the forest) selected so far, this yields a unique directed cycle in the graph $D_M$ defined in the previous section. If not, then there would be a node with two incoming arcs in $D_M$, as every arc on the cycle is oriented in some direction. This would imply that $M$ is not a matching.

Let $e' = \{u, z\} \in E$ be the element corresponding to the incoming arc at $u$ in $D_M$. Note that by assumption $u$ is already matched up, as $e_{\max(v)}$ creates a directed cycle in $D_{M \cup e_{\max(v)}}$. That is, we have $\{e', u\} \in M$. Then, by definition of $\eta$, we have

$$\eta + p_u^* \geq e_{\max}(u) \geq w(u, v) \geq p_v^* - \eta, \tag{5}$$

where the last inequality holds by (3). Combining (5) with the fact that $w(u, z) \geq p_u^* - \lambda$ (because $\{u, z\}$ got matched up to $u$), and the fact that $e_{\max}(v) \geq p_v^* - \eta$, by (3), it follows that

$$2w(u, z) \geq [p_u^* - (\lambda + \eta)] + [p_v^* - (\lambda + \eta)]. \tag{6}$$

In this case, we call $u$ the *ii)-proxy* of $v$.

**Claim 5.4.** *For any distinct $v, v' \in T$, their corresponding proxies $u$ and $u'$ are also distinct.*

*Proof.* Suppose that $u = u'$. The proof proceeds by case distinction based on the proxy types.

1. $u = u'$ is defined as i)-proxy for both $v$ and $v'$: This cannot happen as $u = u'$ would then have been matched up twice by Algorithm 5.

2. $u = u'$ is defined as ii)-proxy for both $v$ and $v'$: In this case there is a directed cycle with the arc $(u, v) = (u', v)$ and another directed cycle with the arc $(u', v') = (u, v')$. Hence, there is a vertex with two incoming arcs in $D_M$. This also means that Algorithm 5 has matched up a vertex twice, which is not possible.

3. $u = u'$ is defined as i)-proxy for $v$ and as ii)-proxy for $v'$: Then $e_{\max}(v)$, which gets matched up with $u = u'$, must have arrived before $e_{\max}(v')$. If not, then both $v'$ and $u = u'$ would have been unmatched when $e_{\max}(v')$ arrived and we could have matched it up with at least $v'$ (as this cannot create a cycle since $u = u'$ is also not matched at that time). This means that when $e_{\max}(v')$ arrived, the reason that we did not match it up to $v'$ is because this would create a directed cycle in $D_{M \cup e_{\max}(v')}$. But, as $u$ has an incoming arc from $v$ in $D_M$, this means that the directed cycle goes through $v$, which implies that $v$ *did* get matched up in Phase II, which we assumed was not the case.

This concludes the proof of the claim. $\qquad\square$

Using Claim 5.2 in combination with (4) and (6), we then find that

$$w[M_{II}] \geq \frac{1}{2} \sum_{v \in Q} [p_v^* - (\lambda + \eta)],$$

where $M_{II}$ contains all the edges obtained in Phase II. Roughly speaking, for every edge $e_{\max}(v)$ that we cannot select in Phase II, there is some other edge selected in Phase II that 'covers' its weight in the summation above (and for every such $v$ we can find a unique edge that has this property).

Now, in general, we have a uniformly random arrival order, and therefore, for every $v \in V$, the probability that edge $e_{\max}(v)$ arrives in Phase II equals $\frac{1}{d} - \frac{1}{c}$. Therefore, with expectation taken over the arrival order, we have

$$\mathbb{E}[M_{II}] \geq \frac{1}{2} \left( \frac{1}{d} - \frac{1}{c} \right) \sum_{v \in V} (p_v^* - (\lambda + \eta)) \geq \frac{1}{2} \left( \frac{1}{d} - \frac{1}{c} \right) \left( 1 - \frac{2(\lambda + \eta)|V|}{\text{OPT}} \right) \text{OPT}.$$

We continue with the worst-case bound that holds even if the prediction error is large. We first analyze the probability that two given nodes $u$ and $v$ do not get matched up in Phase II. Here, we will use the thresholds $t_v$ defined in Algorithm 5.

Conditioned on the set of elements $A$ that arrived in Phase I/II, the probability that the maximum edge weight adjacent to $v$, over all edges adjacent to $v$ in $A$, appears in Phase I is at equal to $(d/c)$. This implies that $v$ will not get matched up in Phase II, by definition of Algorithm 5. The worst-case bound of $(d - 1)/c^2$ is proven in Appendix E. $\qquad\square$

# 6 Conclusion

Our results can be seen as the first evidence that online selection problems are a promising area for the incorporation of machine learned advice following the frameworks of [32, 38]. Many interesting

problems and directions remain open. For example, does there exist a natural prediction model for the general matroid secretary problem? It is still open whether this problem admits a constant-competitive algorithm. Is it possible to show that there exists an algorithm under a natural prediction model that is constant-competitive for accurate predictions, and that is still $O(1/\log(\log(r)))$-competitive in the worst case, matching the results in [13, 28]? Furthermore, although our results are optimal within this specific three phased approach, it remains an open question whether they are optimal in general for the respective problems.

## Footnotes

*Work done in part while the author was at Saarland University and Max-Planck-Institute for Informatics and supported by DFG grant AN 1262/1-1.

[1]This example is similar to the classical secretary problem [15].

[2]To be precise, we consider the so-called *value maximization* version of the problem, see Section 3 for details.

[3]This corresponds to a prediction for the maximum price somebody is willing to offer in the laptop example.

[4]The term "robustness" has also been used in connection with secretary problems (see for example [5]), but in a totally different sense.

[5]Any $\alpha$-approximation for the classical secretary problem yields an $\alpha$-approximation for the value-maximization variant.

[6]Such a prediction does not seem to have any relevance in case the goal is to maximize the probability with which the best secretary is chosen. Intuitively, for every instance $(v_1, \dots, v_n)$ there is a 'more or less isomorphic' instance $(v'_1, \dots, v'_n)$ for which $v'_i < v'_j$ if and only if $v_i < v_j$ for all $i, j$, and for which, all values are close to each other and also to the prediction $p^*$, for which we assume that $p^* < \min_i v'_i$. Any algorithm that uses only pairwise comparisons between the values $v'_i$ can, intuitively, not benefit from the prediction $p^*$. Of course, for the value-maximization variant, choosing any $i$ will be close to optimal in this case if the prediction error is small.

[7]Alternatively, one could define an interval around $p^*$, but given that we get a prediction for the maximum value, this does not make a big difference.

[8]The example given there is for adversarial arrival order, but also applies to uniformly random arrival order.

[9]We call the edges in $E$ (of the original graph $G$) elements, in order to avoid confusion with the edges of $B_G$.

[10]One could have done the same in Algorithm 3, but this leads to a worst-case bound that is worse than $\ln(c/d)/c$.

[11]For the general case, one can use a global ordering on all edges in $E$ and break ties where needed.

[12] We use notation and terminology closely related to that in [4].

[13] We use $\rho_i$ in order to avoid confusion with the predictions $p_r^*$.

[14] This submatching corresponds to the online matching $M$ that we construct in Algorithm 3.

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

# A   Secretary Problem

This section is organized as follows:

In Subsection A.1, we show that the discontinuity at $\eta = \lambda$, in Theorem 3.1, can be smoothed out by selecting $\lambda$ according to some distribution with mean representing our confidence in the prediction $p^*$. Further, we study the competitive ratio as a function of the prediction error $\eta$.

In Subsection A.2, we demonstrate that our deterministic Algorithm 1 can achieve a better competitive ratio than its corresponding naive randomization.

## A.1   Randomized Algorithms

We note that Algorithm 1 although deterministic, can be relatively easily transformed to an algorithm that picks the confidence parameter $\lambda \in [0, p^*]$ according to some probability distribution. Algorithm 1 is then the special case where the whole probability mass of the distribution is at one point in $[0, p^*]$.

This naturally gives rise to the question of whether there exists a distribution that outperforms the deterministic algorithm. It can be relatively easily seen, that the deterministic algorithm with $\lambda = \eta$ is the best possible competitive ratio that can be obtained with such an approach. Therefore, a randomized algorithm can at best match the deterministic one with $\lambda = \eta$, and this happens only in the case in the center of mass of the used distribution is at $\eta$. Since $\eta$ is unknown to the algorithm, it is not possible to select a distribution that outperforms any deterministic algorithm for all possible $\eta$'s.

Despite that, it may still be advantageous to pick $\lambda$ at random according to some distribution, in order to avoid the "jump" that occurs at $\eta = \lambda$ in the competitive ratio of the deterministic algorithm. In particular for an appropriate distribution with density function $h_\lambda(x)$ the expected competitive ratio is given by:

$$\mathbb{E}_\lambda\left[g_{c,\lambda}(\eta)\right] = Pr[\lambda < \eta] \cdot \frac{1}{ce} + f(c) \int_\eta^{p^*} h_\lambda(x)\left(1 - \frac{x + \eta}{OPT}\right) dx,$$

which can be seen as a convex combination of two competitive ratios.

Some example distributions and how they compare to an algorithm that selects $\lambda$ deterministically can be seen in Figure 2.

## A.2   Comparison between Algorithm 1 and its naive randomization

One natural question that arises from the bound in Theorem 3.1 is whether one can significantly improve the result using randomization.

Here, we provide a brief comparison with the following naive randomization of Algorithm 1, which randomly chooses between running the classical secretary problem without predictions and (roughly speaking) the greedy prediction-based procedure in Phase II in Algorithm 1. That is, given $\gamma \in [0, 1]$, with probability $\gamma$ it runs the classical secretary problem, and with probability $1 - \gamma$, it runs the prediction-based algorithm that simply selects the first element with value greater or equal than $p^* - \lambda$ (if any). Note that its expected competitive ratio at least

$$\gamma \frac{1}{e} + (1 - \gamma)\left(\max\left\{1 - \frac{\lambda + \eta}{\text{OPT}}, 0\right\}\right). \tag{7}$$

(a)　　　　　　　　　　　　　　　　　　　　(b)

(c)

Figure 2: A comparison the deterministic (in red) and randomized (in blue) choice of $\lambda$. The $y$-axes are the competitive ratio, the $x$-axes are the prediction error $\eta$, and all figures consider $p^* = 100$. In Subfigure 2a we take $\lambda = 25$ for the deterministic algorithm and choose $\lambda$ according to the uniform distribution in $(20, 30)$ for the randomized one. In Subfigure 2b we have $\lambda \approx 25$ for the deterministic one and the normal distribution with mean 25 and variance 10. Finally, in Subfigure 2c we have $\lambda \approx 25$, and the normal distribution with mean 0 and variance 32. All plots have $c = 2$.

In order to compare Algorithm 1 with its naive randomization, we set $\gamma = 1/c$. This implies that both algorithms are at least $1/(ce)$-competitive in the worst-case when the predictions are poor. Having the same worst-case guarantee, we now focus on their performance in case when the prediction error $\eta$ and the confidence parameter $\lambda$ are small. In particular, let us consider the case where $\lambda + \eta = \delta \cdot \text{OPT}$ for some small $\delta > 0$. Then, the expected competitive ratio in (7) reduces to

$$\frac{1}{ce} + \left(1 - \frac{1}{c}\right)(1 - \delta). \tag{8}$$

We now compare the expected competitive ratio of Algorithm 1 and its naive randomization, which read $f(c)(1 - \delta)$ and (8) respectively. In Figure 3, we conduct a numerical experiment with fixed $\delta = 0.1$ and $\lambda + \eta = 0.1\text{OPT}$. Our experimental data indicates that for $c \geq 1.185$, Algorithm 1 is at least $1/e$-competitive and it significantly outperforms the classical secretary algorithm as $c$ increases. Furthermore, for $c \geq 1.605$ Algorithm 1 performs better than its naive randomization. On the other hand, we note that our experiments indicate that as $\delta$ increases the competitive advantage of Algorithm 1 over its naive randomization decreases.

Figure 3: The black horizontal line indicates the tight bound of $1/e$ for the classical secretary algorithm. The bold blue line is the performance guarantee for Algorithm 1; and the dashed red line is the performance guarantee for the obvious randomized algorithm.

## B  Perfect Matching Instances

Given an undirected weighted bipartite graph $G' = (L' \cup R', E', w')$ we construct an augmented bipartite graph $G = (L \cup R, E, w)$ as follows:

i) the left node set $L = L'$; ii) the right node set $R = R' \cup L'$; iii) the edge set $E = E' \cup F'$ where the set $F'$ consists of edges $\{u_i, v_i\}$ such that $u_i$ and $v_i$ are the $i$-th node in $L$ and $L'$ respectively, for all $i \in \{1, \ldots, |L|\}$; iv) $w(e) = w'(e)$ for all edges $e \in E'$ and $w(e) = 0$ for all edges $e \in F'$.

We call the resulting bipartite graph $G$ *perfect*.

**Fact B.1.** *Suppose $G = (L \cup R, E, w)$ is a perfect bipartite graph. Let $\ell \in \{1, \ldots, |L|\}$ be an arbitrary index and $\mathcal{L}(\ell) = \{S \subseteq L : |S| = \ell\}$ be the set of all subsets of nodes in $L$ of size $\ell$. Then, for every subset $S \in \mathcal{L}(\ell)$ the induced subgraph $G[S \cup N(S)]$ has a perfect matching $M_S$ of size $\ell$, i.e., $|M_S| = |S| = \ell$.*

# C  General analysis of the algorithm of Kesselheim et al. [22]

In this section, we analyze a modified version of the algorithm of Kesselheim et al. [22], see Algorithm 6. Our analysis extends the proof techniques presented in [22, Lemma 1].

**Theorem C.1.** *Given a perfect bipartite graph, Algorithm 6 is $(\frac{1}{c} - \frac{1}{n}) \ln \frac{c}{d}$ competitive in expectation. In addition, the expected weighted contribution of the nodes $\{\lfloor n/d \rfloor + 1, \ldots, n\}$ to the online matching $M$ is $OPT \cdot (\frac{1}{c} - \frac{1}{n}) \ln d$.*

For convenience of notation, we will number the vertices in $L$ from 1 to $n$ in the random order they are presented to the algorithm. Hence, we will use the variable $\ell$ as an integer, the name of an iteration and the name of the current node (the last so far).

---

**ALGORITHM 6:** Online bipartite matching algorithm (under uniformly random vertex arrivals)

---

**Input**  : Vertex set $R$ and cardinality $|L| = n$.
**Output:** Matching $M$.

**Phase I:**
**for** $\ell = 1, \ldots, \lfloor n/c \rfloor$ **do**
  | Observe arrival of node $\ell$, but do nothing.
**end**
Let $L' = \{1, \ldots, \lfloor n/c \rfloor\}$ and $M = \emptyset$.
**Phase II:**
**for** $\ell = \lfloor n/c \rfloor + 1, \ldots, \lfloor n/d \rfloor$ **do**
  | Let $L' = L' \cup \ell$.
  | Let $M^{(\ell)} =$ optimal matching on $G[L' \cup R]$.
  | Let $e^{(\ell)} = (\ell, r)$ be the edge assigned to $\ell$ in $M^{(\ell)}$.
  | **if** $M \cup e^{(\ell)}$ *is a matching* **then**
  |   | Set $M = M \cup \{e^{(\ell)}\}$.
  | **end**
**end**

---

### Organization

In Subsection C.1, we present the notation. In Subsection C.2, we give the main structural result and prove Theorem C.1. In addition, in Subsection C.3, we give a lower bound on the probability that an arbitrary node $r \in R$ remains unmatched after the completion of Phase II.

## C.1  Notation

Consider the following random process:

Sample uniformly at random a permutation of the nodes $L$. Let $L_\ell$ be a list containing the first $\ell$ nodes in $L$, in the order as they appear, and let $M^{(\ell)}$ be the corresponding optimum matching of the induced subgraph $G^{(\ell)}$ on the node set $L_\ell \cup N(L_\ell)$.

Let $E^{(\ell)}$ be the event $\{e^{(\ell)} \cup M$ is a matching$\}$, where (r.v.) $M$ is the current online matching. Note that the existence of edge (r.v.) $e^{(\ell)}$ is guaranteed by the Fact B.1 and $G$ is a perfect bipartite graph. We define a random variable

$$A_\ell = \begin{cases} w(e^{(\ell)}) & \text{, if event } E^{(\ell)} \text{ occur;} \\ 0 & \text{, otherwise.} \end{cases}$$

## C.2  Structural Lemma

**Lemma C.2.** *Suppose $G = (L \cup R, E, w)$ is a perfect bipartite graph. Then, for every $c > 1$ it holds for every $\ell \in \{\lfloor n/c \rfloor + 1, \ldots, n\}$ that*

$$\mathbb{E}[A_\ell] \geq \frac{\lfloor n/c \rfloor}{n} \cdot \frac{OPT}{\ell - 1}.$$

Before we prove Lemma C.2, we show that it implies Theorem C.1.

### C.2.1  Proof of Theorem C.1

Using Lemma C.2, we have

$$\mathbb{E}\left[\sum_{\ell=1}^{n/d} A_\ell\right] = \sum_{\ell=\lfloor n/c \rfloor + 1}^{n/d} \mathbb{E}\left[A_\ell\right] \geq \sum_{\ell=\lfloor n/c \rfloor + 1}^{n/d} \frac{\lfloor n/c \rfloor}{n} \cdot \frac{OPT}{\ell - 1} \geq OPT \cdot \left(\frac{1}{c} - \frac{1}{n}\right) \cdot \ln\frac{c}{d},$$

where the inequalities follow by combining $\frac{\lfloor n/c \rfloor}{n} \geq \frac{1}{c} - \frac{1}{n}$ and

$$\sum_{\ell=\lfloor n/c \rfloor + 1}^{n/d} \frac{1}{\ell - 1} = \sum_{\ell=\lfloor n/c \rfloor}^{n/d-1} \frac{1}{\ell} \geq \ln\frac{n/d}{\lfloor n/c \rfloor} \geq \ln\frac{c}{d}.$$

### C.2.2  Proof of Lemma C.2

We prove Lemma C.2 in two steps. Observe that $\mathbb{E}[A_\ell \mid \neg E^{(\ell)}] = 0$ implies

$$\mathbb{E}\left[A_\ell\right] = \mathbb{E}\left[w(e^{(\ell)}) \mid E^{(\ell)}\right] \cdot Pr\left[E^{(\ell)}\right].$$

We proceed by showing, in Lemma C.3, that $\mathbb{E}\left[w(e^{(\ell)}) \mid E^{(\ell)}\right] \geq \frac{OPT}{n}$, and then in Lemma C.4 that $Pr\left[E^{(\ell)} \mid E^{(\ell)}\right] \geq \frac{\lfloor n/c \rfloor}{\ell - 1}$.

Let $S$ be a subset of $L$ of size $\ell$, and let $M_S$ be the optimum weighted matching w.r.t. the induced subgraph $G[S \cup N(S)]$. For a fixed subset $S \subseteq L$ with size $\ell$, let $R_\ell(S)$ be the event that {the node set of $L_\ell$ equals $S$}, i.e. $\text{Set}(L_\ell) = S$. Let $\mathcal{L}(\ell)$ be the set of all subsets of $L$ of size $\ell$, i.e., $\mathcal{L}(\ell) = \{S \subseteq L : |S| = \ell\}$.

**Lemma C.3.** *For every perfect bipartite graph $G = (L \cup R, E, w)$ it holds that*

$$\mathbb{E}\left[w(e^{(\ell)}) \mid E^{(\ell)}\right] \geq \frac{OPT}{n}.$$

*Proof.* Using conditional expectation,

$$\mathbb{E}\left[w(e^{(\ell)}) \mid E^{(\ell)}\right] = \sum_{S \in \mathcal{L}(\ell)} \mathbb{E}\left[w(e^{(\ell)}) \mid R_\ell(S) \wedge E^{(\ell)}\right] \cdot Pr\left[R_\ell(S)\right]. \tag{9}$$

Since the order of $L$ is sampled u.a.r. we have $Pr\left[R_\ell(S)\right] = 1/\binom{n}{\ell}$, and thus it suffices to focus on the conditional expectation

$$\mathbb{E}\left[w(e^{(\ell)}) \mid R_\ell(S) \wedge E^{(\ell)}\right] = \sum_{e^{(i)} \in M_S} w(e^{(i)}) \cdot Pr_{e^{(\ell)} \sim M_S}\left[e^{(\ell)} = e^{(i)}\right]$$

$$= \frac{1}{\ell} \sum_{e^{(i)} \in M_S} w(e^{(i)}). \tag{10}$$

where the last equality uses $G$ is a perfect bipartite graph and Fact B.1. Then, by combining (9,10) we have

$$\mathbb{E}\left[w(e^{(\ell)}) \mid E^{(\ell)}\right] = \frac{1}{\binom{n}{\ell}} \cdot \frac{1}{\ell} \sum_{S \in \mathcal{L}(\ell)} \sum_{e^{(i)} \in M_S} w(e^{(i)}). \tag{11}$$

Observe that for any subset $S \subseteq L$, it holds for $M^\star|_S = \{e^{(i)} = (i, r_i) \in M^\star : i \in S\}$ the restriction of the optimum matching $M^\star$ (w.r.t. the whole graph $G$) on $S$ that

$$\sum_{e^{(i)} \in M_S} w(e^{(i)}) \geq \sum_{e^{(i)} \in M^\star|_S} w(e^{(i)}). \tag{12}$$

Further, since every vertex $i \in L(M^\star)$ appears in $\binom{n-1}{\ell-1}$ many subsets of size $\ell$ and $\binom{n-1}{\ell-1}/\binom{n}{\ell} = \ell/n$, it follows by (11,12) that

$$\begin{aligned}
\mathbb{E}\left[A_\ell \mid E^{(\ell)}\right] &\geq \frac{1}{\binom{n}{\ell}} \cdot \frac{1}{\ell} \sum_{S \in \mathcal{L}(\ell)} \sum_{e^{(i)} \in M^\star|_S} w(e^{(i)}) \\
&= \frac{\binom{n-1}{\ell-1}}{\binom{n}{\ell}} \cdot \frac{1}{\ell} \sum_{e^{(i)} \in M^\star} w(e^{(i)}) = \frac{OPT}{n}.
\end{aligned}$$

$\square$

**Lemma C.4.** *For every perfect bipartite graph $G = (L \cup R, E, w)$ it holds that*

$$Pr\left[E^{(\ell)}\right] \geq \frac{\lfloor n/c \rfloor}{\ell - 1}. \tag{13}$$

*Proof.* For a fixed subset $S \subseteq L$ with size $\ell$, let $F_{S,\ell}^{(i)}$ be the event that {event $R_\ell(S)$ occurs} *and* {the edge $e^{(\ell)} = (i, r_i)$}. Then, by conditioning on the choice of subset $S \in \mathcal{L}(\ell)$, we have

$$\begin{aligned}
Pr\left[E^{(\ell)}\right] &= \frac{1}{\binom{n}{\ell}} \sum_{S \in \mathcal{L}(\ell)} Pr\left[e^{(\ell)} \cup M \text{ is a matching} \mid R_\ell(S)\right] \\
&= \frac{1}{\binom{n}{\ell}} \frac{1}{\ell} \sum_{S \in \mathcal{L}(\ell)} \sum_{(i,r_i) \in M_S} Pr\left[(i, r_i) \cup M \text{ is a matching} \mid F_{S,\ell}^{(i)}\right]. \tag{14}
\end{aligned}$$

*Note that in (14), the subset $S \subseteq L$ with size $\ell$ and the edge $(i, r_i) \in M_S$ are fixed!*

Given the first $k$ nodes of $L$, in the order as they arrive, and the corresponding perfect matching (r.v.) $M^{(k)}$, let (r.v.) $e^{(k)} = (k, r_k)$ be the edge matched in (r.v.) $M^{(k)}$ from the last node (r.v.) $k$. Note that since $G$ is perfect, it follows by Fact B.1 that edge $e^{(k)}$ exists and $|M^{(k)}| = k$. We denote by (r.v.) $M^{(k)}[k]$ the corresponding right node $r_k$.

Let $Q_k$ be the event that

$$\{\text{node } r_i \notin M^{(k)}\} \vee \left\{\{\text{node } r_i \in M^{(k)}\} \wedge \{M^{(k)}[k] \neq r_i\}\right\}.$$

Then, we have

$$Pr\left[(i, r_i) \cup M \text{ is a matching} \mid F_{S,\ell}^{(i)}\right] = Pr\left[\bigwedge_{k=\lfloor n/c \rfloor+1}^{\ell-1} Q_k \mid F_{S,\ell}^{(i)}\right]. \tag{15}$$

Observe that the probability of event $\vdash Q_k$ is equal to

$$Pr\left[\{\text{node } r_i \in M^{(k)}\} \wedge \left\{\{\text{node } r_i \notin M^{(k)}\} \vee \{M^{(k)}[k] = r_i\}\right\}\right]$$

$$= Pr\left[\{\text{node } r_i \in M^{(k)}\} \wedge \{M^{(k)}[k] = r_i\}\right]. \tag{16}$$

Let $W_{i,t}$ be the event that $F_{S,\ell}^{(i)} \wedge \left(\bigwedge_{j=k}^{t-1} Q_j\right)$ for $t \in \{k+1, \ldots, \ell-1\}$, and $W_{i,k}$ be the event $F_{S,\ell}^{(i)}$. Using conditional probability, we have

$$Pr\left[\bigwedge_{k=\lfloor n/c \rfloor+1}^{\ell-1} Q_k \mid F_{S,\ell}^{(i)}\right] = Pr\left[Q_{\ell-1} \mid W_{i,\ell-1}\right] \cdots Pr\left[Q_{k+1} \mid W_{i,k+1}\right] \cdot Pr\left[Q_k \mid W_{i,k}\right]. \tag{17}$$

We now analyze the terms in (17) separately. Let $\mathcal{T}(i,k)$ be the set of all matchings $M^{(k)}$ satisfying the event $F_{S,\ell}^{(i)} \wedge \{\text{node } r_i \in M^{(k)}\}$. Using (16), we have

$$Pr\left[\vdash Q_k \mid F_{S,\ell}^{(i)}\right] = Pr\left[\{\text{node } r_i \in M^{(k)}\} \wedge \{M^{(k)}[k] = r_i\} \mid F_{S,\ell}^{(i)}\right]$$

$$\leq Pr\left[M^{(k)}[k] = r_i \mid F_{S,\ell}^{(i)} \wedge \{\text{node } r_i \in M^{(k)}\}\right]$$

$$= \frac{1}{|\mathcal{T}(i,k)|} \sum_{M' \in \mathcal{T}(i,k)} \frac{1}{|M'|} = \frac{1}{k}, \tag{18}$$

where the last equality follows by Fact B.1 and $G$ is perfect. Thus,

$$Pr\left[Q_k \mid W_{i,k}\right] = 1 - Pr\left[\vdash Q_k \mid F_{S,\ell}^{(i)}\right] \geq 1 - \frac{1}{k}. \tag{19}$$

Similarly, for $t \in \{k+1, \ldots, \ell-1\}$ we have

$$Pr\left[\vdash Q_t \mid W_{i,t}\right] \leq Pr\left[M^{(t)}[t] = r_i \mid W_{i,t} \wedge \{\text{node } r_i \in M^{(t)}\}\right] = \frac{1}{t},$$

and therefore

$$Pr\left[Q_t \mid W_{i,t}\right] = 1 - Pr\left[\vdash Q_t \mid W_{i,t}\right] \geq 1 - \frac{1}{t}. \tag{20}$$

By combining (15,17,19,20), we obtain

$$Pr\left[(i, r_i) \cup M \text{ is a matching} \mid F_{S,\ell}^{(i)}\right] \geq \prod_{k=\lfloor n/c \rfloor+1}^{\ell-1} \left(1 - \frac{1}{k}\right) = \frac{\lfloor n/c \rfloor}{\ell - 1}. \tag{21}$$

Since every summand in (14) is lower bounded by (21), we have

$$Pr\left[E^{(\ell)} \mid E_{\exists}^{(\ell)}\right] \geq \frac{\lfloor n/c \rfloor}{\ell - 1}.$$

$\square$

## C.3  Algorithm 3 (Omitted Proofs)

We now lower bound the probability that an arbitrary node $r \in R$ remains unmatched after the completion of Phase II in Algorithm 3. Our analysis uses similar arguments as in Lemma C.4, but for the sake of completeness we present the proof below.

**Lemma C.5.** *For every constants $c \geq d \geq 1$ and for every perfect bipartite graph $G = (L \cup R, E, w)$, it holds for every node $r \in R$ that*

$$Pr\left[r \text{ is not matched in Phase II}\right] \geq \frac{d}{c} - o(1).$$

*Proof.* Observe that

$$Pr\left[r \text{ is not matched in Phase II}\right] = Pr\left[\bigwedge_{k=\lfloor n/c \rfloor +1}^{\lfloor n/d \rfloor} Q_k\right].$$

Using (21), we have

$$Pr\left[r \text{ is not matched in Phase II}\right] \geq \prod_{k=\lfloor n/c \rfloor +1}^{\lfloor n/d \rfloor} \left(1 - \frac{1}{k}\right) \geq \frac{\lfloor n/c \rfloor}{\lfloor n/d \rfloor} \geq \frac{d}{c} - \frac{d}{n}.$$

□

# D   Deterministic Graphic Matroid Secretary Algorithm

In this section, we analyze the competitive ratio of Algorithm 4.

**Theorem 5.2.** *The deterministic Algorithm 4 is $(1/4 - o(1))$-competitive for the graphic matroid secretary problem.*

The rest of this section is devoted to proving Theorem 5.2, and is organized as follows. In Subsection D.1, we give two useful summation closed forms. In Subsection D.2, we present our notation. In Subsection D.3, we extend Lemma C.2 to bipartite-matroid graphs. In Subsection D.4, we prove Theorem 5.2.

## D.1   Summation bounds

**Claim D.1.** *For any $k \in \mathbb{N}$ and $n \in \mathbb{N}_+$ , we have*

$$\sum_{\ell=n}^{n+k} \frac{1}{\ell \cdot (\ell+1)} = \frac{k+1}{n(n+k+1)}.$$

*Proof.* The proof is by induction. The base case follows by

$$\frac{1}{n} \cdot \frac{1}{n+1} + \frac{1}{n+1} \cdot \frac{1}{n+2} = \frac{2}{n(n+2)}.$$

Our inductive hypothesis is $\sum_{\ell=n}^{n+k} \frac{1}{\ell \cdot (\ell+1)} = \frac{k+1}{n(n+k+1)}$. Then, we have

$$\sum_{\ell=n}^{n+k+1} \frac{1}{\ell} \cdot \frac{1}{\ell+1} = \frac{k+1}{n(n+k+1)} + \frac{1}{n+k+1} \cdot \frac{1}{n+k+2} = \frac{1}{n+k+1} \left[ \frac{k+1}{n} + \frac{1}{n+k+2} \right]$$

$$= \frac{1}{n+k+1} \left[ \frac{(k+1)(n+k+1) + n + k + 1}{n(n+k+2)} \right] = \frac{k+2}{n(n+k+2)}.$$

$\square$

**Claim D.2.** *For any $c > 1$, it holds that*

$$f(c,n) := \frac{1}{n} \sum_{\ell=\lfloor n/c \rfloor}^{n-1} \frac{(\lfloor n/c \rfloor - 1) \lfloor n/c \rfloor}{(\ell-1)\ell} = \frac{\lfloor n/c \rfloor}{n} \cdot \left[ \frac{n-1}{n-2} - \frac{\lfloor n/c \rfloor}{n-2} \right] \gtrsim \frac{c-1}{c^2}.$$

*In particular, the lower bound is maximized for $c = 2$ and yields $f(2,n) \gtrsim 1/4$.*

*Proof.* By Claim D.1, we have

$$\sum_{\ell=\lfloor n/c \rfloor}^{n-1} \frac{1}{\ell-1} \cdot \frac{1}{\ell} = \sum_{\ell=\lfloor n/c \rfloor - 1}^{\lfloor n/c \rfloor + [n - \lfloor n/c \rfloor - 2]} \frac{1}{\ell} \cdot \frac{1}{\ell+1} = \frac{n - \lfloor n/c \rfloor - 1}{(\lfloor n/c \rfloor - 1)(n-2)},$$

and thus

$$\frac{1}{n} \sum_{\ell=\lfloor n/c \rfloor}^{n-1} \frac{(\lfloor n/c \rfloor - 1) \lfloor n/c \rfloor}{(\ell-1)\ell} = \frac{(\lfloor n/c \rfloor - 1) \lfloor n/c \rfloor}{n} \cdot \frac{n - \lfloor n/c \rfloor - 1}{(\lfloor n/c \rfloor - 1)(n-2)}$$

$$= \frac{\lfloor n/c \rfloor}{n} \cdot \left[ \frac{n-1}{n-2} - \frac{\lfloor n/c \rfloor}{n-2} \right]$$

$$\geq \left( \frac{1}{c} - \frac{1}{n} \right) \cdot \left[ 1 + \frac{1}{n-2} - \frac{n}{n-2} \cdot \frac{1}{c} \right]$$

$$\gtrsim \frac{1}{c} \cdot \left[ 1 - \frac{1}{c} \right] = \frac{c-1}{c^2}.$$

Let $g(x) = (x-1)/x^2$. Observe that its first derivative satisfies

$$\frac{d}{dx}g(x) = \frac{x^2 - (x-1)2x}{x^4} = \frac{x(2-x)}{x^4} = 0 \iff x_1 = 0 \quad x_2 = 2.$$

Further, $g(x)$ decreases in the range $[-\infty, 0]$, increases in $[0, 2]$ and again decreases in $[2, \infty]$. Hence, we have $\max_{x>0} g(x) = g(2) = 1/4$. $\qquad\square$

## D.2  Notation

Given an undirected weighted graph $G' = (V, E', w')$, we construct a bipartite-matroid graph $G = (L \cup R, E, w)$ as follows:

i) let the set of the right nodes be $R = V$; ii) the set of the left nodes be $L = E'$, i.e., $\{u, v\} \in L$ if $\{u, v\} \in E'$; iii) and for each edge in $\{u, v\} \in E'$ we insert two edges $\{\{u, v\}, u\}, \{\{u, v\}, v\} \in E$ with equal weight $w(\{\{u, v\}, u\}) = w(\{\{u, v\}, v\}) = w'(\{u, v\})$.

Although $M^{(\ell)}$ is a matching, we note that $M$ is not a matchings in the strict sense, since for an edge $\{\{u, v\}, u\} \in E$ (similarly $\{\{u, v\}, v\} \in E$) to be matched it is required that both nodes $u, v \in R$ are not yet matched. To emphasize this, we refer to $M$ as "matching$^\star$".

## D.3  Structural Lemma

We now extend Lemma C.2 to bipartite-matroid graphs.

**Lemma D.3.** *Suppose $G = (L \cup R, E, w)$ is a perfect bipartite-matroid graph as above. Then, for every $c > 1$ it holds for every $\ell \in \{\lfloor m/c \rfloor + 1, \ldots, m\}$ that*

$$\mathbb{E}[A_\ell] \geq \frac{\lfloor m/c \rfloor - 1}{(\ell - 1) - 1} \cdot \frac{\lfloor m/c \rfloor}{\ell - 1} \cdot \frac{OPT}{m}.$$

It is straightforward to verify that the statement of Lemma C.3 holds for perfect bipartite-matroid graphs, and yields

$$\mathbb{E}\left[w(e^{(\ell)}) \mid E^{(\ell)}\right] \geq \frac{OPT}{m}.$$

Hence, to prove Lemma D.3 it remains to extend the statement of Lemma C.4.

**Lemma D.4.** *For every perfect bipartite-matroid graph $G = (L \cup R, E, w)$ it holds that*

$$Pr\left[E^{(\ell)}\right] \geq \frac{\lfloor m/c \rfloor - 1}{(\ell - 1) - 1} \cdot \frac{\lfloor m/c \rfloor}{\ell - 1}.$$

*Proof.* We follow the proof in Lemma C.4, with the amendment that a node $\{u_k, v_k\} \in L$ and an edge $\{\{u_k, v_k\}, r_k\} \in E$. Recall that for a *fixed* subset $S \subseteq L$ with size $\ell$ and an edge $(i, r_i) \in M_S$, we can condition on the event $F_{S,\ell}^{(i)}$ that {the node set of $L_\ell$ equals $S$} *and* {the edge $e^{(\ell)} = (i, r_i)$}.

Let $Q_k^r$ be the event that

$$\{\text{node } r \notin M^{(k)}\} \vee \left\{\{\text{node } r \in M^{(k)}\} \wedge \{M^{(k)}[k] \neq r\}\right\},$$

and let $\mathcal{P}_k$ be the event that $Q_k^{u_i} \wedge Q_k^{v_i}$. Then, we have

$$Pr\left[\{\{u_i, v_i\}, r_i\} \cup M \text{ is a matching}^\star \mid F_{S,\ell}^{(i)}\right] = Pr\left[\bigwedge_{k=\lfloor m/c \rfloor + 1}^{\ell - 1} \mathcal{P}_k \mid F_{S,\ell}^{(i)}\right].$$

Combining the Union bound and (18), yields

$$
\begin{aligned}
Pr\left[\neg\,\mathcal{P}_k \mid F_{S,\ell}^{(i)}\right] &= Pr\left[\neg\,Q_k^{u_i} \vee \neg\,Q_k^{v_i} \mid F_{S,\ell}^{(i)}\right] \\
&\leq Pr\left[\neg\,Q_k^{u_i} \mid F_{S,\ell}^{(i)}\right] + Pr\left[\neg\,Q_k^{v_i} \mid F_{S,\ell}^{(i)}\right] \\
&\leq \frac{2}{k}.
\end{aligned}
$$

Hence, using similar arguments as in the proof in Lemma C.4, we have

$$
Pr\left[E^{(\ell)}\right] \geq \prod_{k=\lfloor m/c \rfloor + 1}^{\ell - 1} \left(1 - \frac{2}{k}\right) = \frac{\lfloor m/c \rfloor - 1}{(\ell - 1) - 1} \cdot \frac{\lfloor m/c \rfloor}{\ell - 1}.
$$

$\square$

## D.4  Proof of Theorem 5.2

Using Lemma D.3, we have

$$
\mathbb{E}\left[\sum_{\ell=1}^{m} A_\ell\right] = \sum_{\ell=\lfloor m/c \rfloor + 1}^{m} \mathbb{E}\left[A_\ell\right] \geq \frac{OPT}{m} \sum_{\ell=\lfloor m/c \rfloor + 1}^{m} \frac{\lfloor m/c \rfloor - 1}{(\ell - 1) - 1} \cdot \frac{\lfloor m/c \rfloor}{\ell - 1}.
$$

The statement follows by Claim D.2 and noting that

$$
\frac{1}{m} \sum_{\ell=\lfloor m/c \rfloor + 1}^{m} \frac{\lfloor m/c \rfloor - 1}{(\ell - 1) - 1} \cdot \frac{\lfloor m/c \rfloor}{\ell - 1} \geq \left(\frac{c - 1}{c^2} - o(1)\right) \geq \left(\frac{1}{4} - o(1)\right). \tag{22}
$$

# E  Graphic Matroid Secretary Algorithm with Predictions

In this section, we prove the worst-case bound of $(d-1)/c^2$ in Theorem 5.3, by analyzing Phase III of Algorithm 5.

**Theorem E.1.** *The Phase III in Algorithm 5 is $(\frac{d-1}{c^2} - o(1))$-competitive.*

The rest of this section is denoted to proving Theorem E.1, and is organized as follows. In Subsection E.1, we analyze the probability that a fixed pair of distinct vertices is eligible for matching in Phase III. In Subsection E.2, we give a lower bound on the event that $\{e^{(\ell)} \cup M$ is a matching$^\star\}$. In Subsection E.3, we prove Theorem E.1.

## E.1  Pairwise Node Eligibility in Phase III

For any distinct nodes $u, v \in R$, we denote by $\Phi_{u,v}^{\notin M}$ the event that

$$\{u \text{ and } v \text{ are } not \text{ matched in Phase II}\}.$$

**Claim E.2.** *It holds that*

$$Pr\left[\Phi_{u,v}^{\notin M}\right] \geq \left(\frac{d}{c}\right)^2 \cdot \frac{1 - \frac{c}{m}}{1 - \frac{d}{m}}.$$

*Proof.* Let $S$ be a random variable denoting the set of all nodes in $L$ that appear in Phase I and Phase II. Let

$$e'_{\max}(u, S) = \arg \max_{\{u,z\} \in S} w(u, z)$$

be a random variable denoting the node $\{u, z\} \in L$ with largest weight seen in the set $S$.

The proof proceeds by case distinction:

**Case 1.** Suppose $e'_{\max}(u, S) = e'_{\max}(v, S)$, i.e., there is a node $\{u, v\} \in S$. Let $\mathcal{K}_r(S)$ be the event that node $e'_{\max}(r, S) \in S$ is sampled in Phase I. By conditioning on the choice of $S$, we have

$$Pr\left[\Phi_{u,v}^{\notin M}\right] = \sum_S Pr\left[\mathcal{K}_r(S) \mid S\right] \cdot Pr\left[S\right] = \frac{m/c}{m/d} \cdot \sum_S Pr\left[S\right] = \frac{d}{c}.$$

**Case 2.** Suppose $e'_{\max}(u, S) \neq e'_{\max}(v, S)$, i.e., there are distinct nodes $\{u, x\} \in S$ and $\{v, y\} \in S$ with largest weight, respectively from $u$ and $v$. By conditioning on the choice of $S$, we have

$$Pr\left[\mathcal{K}_u(S) \wedge \mathcal{K}_v(S) \mid S\right] = \frac{2\binom{m}{2} \cdot (\frac{m}{d} - 2)!}{(\frac{m}{d})!} = \left(\frac{d}{c}\right)^2 \cdot \frac{1 - \frac{c}{m}}{1 - \frac{d}{m}},$$

and thus

$$Pr\left[\Phi_{u,v}^{\notin M}\right] = \sum_S Pr\left[\mathcal{K}_u(S) \wedge \mathcal{K}_v(S) \mid S\right] \cdot Pr\left[S\right] = \left(\frac{d}{c}\right)^2 \cdot \frac{1 - \frac{c}{m}}{1 - \frac{d}{m}}.$$

$\square$

## E.2  Lower Bounding the Matching$^\star$ Event

Recall that $E^{(\ell)}$ denotes the event that $\{e^{(\ell)} \cup M$ is a matching$^\star\}$, where (r.v.) $M$ is the current online matching$^\star$, see Subsection D.2 for details. In order to control the possible negative side effect of selecting suboptimal edges in Phase II, we extend Lemma D.4 and give a lower bound on the event $E^{(\ell)}$ for any node $\ell \in L$ that appears in Phase III.

**Lemma E.3.** *For every perfect bipartite-matroid graph $G = (L \cup R, E, w)$, Algorithm 5 guarantees in Phase III that*

$$Pr\left[E^{(\ell)}\right] \geq \frac{\lfloor m/d \rfloor - 1}{(\ell - 1) - 1} \cdot \frac{\lfloor m/d \rfloor}{\ell - 1} \cdot Pr\left[\Phi_{u,v}^{\notin M}\right], \quad \forall \ell \in \{\lfloor m/d \rfloor + 1, \ldots, m\}.$$

*Proof.* We follow the proof in Lemma C.4, with the amendment that a node $\{u_k, v_k\} \in L$ and an edge $\{\{u_k, v_k\}, r_k\} \in E$. Recall that for a *fixed* subset $S \subseteq L$ with size $\ell$ and a *fixed* edge $\{\{u_i, v_i\}, r_i\} \in M_S$, we can condition on the event $F_{S,\ell}^{(i)}$ that

$$\{\text{the set of nodes of } L_\ell \text{ equals } S\} \text{ and } \{\text{the edge } e^{(\ell)} = \{\{u_i, v_i\}, r_i\}\}.$$

Let $Q_k^r$ be the event that

$$\{\text{node } r \notin M^{(k)}\} \vee \left\{\{\text{node } r \in M^{(k)}\} \wedge \{M^{(k)}[k] \neq r\}\right\},$$

and let $\mathcal{P}_k$ denotes the event $Q_k^{u_i} \wedge Q_k^{v_i}$. The proof proceeds by case distinction:

**Case 1.** For $\ell = \lfloor m/d \rfloor + 1$, we have

$$Pr\left[E^{(\ell)}\right] = Pr\left[\{\{u_i, v_i\}, r_i\} \cup M \text{ is a matching}^\star \mid F_{S,\ell}^{(i)}\right] = Pr\left[\Phi_{u,v}^{\notin M}\right].$$

**Case 2.** For $\ell = \{\lfloor m/d \rfloor + 2, \ldots, m\}$, we have

$$\begin{aligned}
Pr\left[E^{(\ell)}\right] &= Pr\left[\{\{u_i, v_i\}, r_i\} \cup M \text{ is a matching}^\star \mid F_{S,\ell}^{(i)}\right] \\
&= Pr\left[\bigwedge_{k=\lfloor m/d \rfloor + 1}^{\ell-1} \mathcal{P}_k \mid F_{S,\ell}^{(i)} \wedge \Phi_{u,v}^{\notin M}\right] \cdot Pr\left[\Phi_{u,v}^{\notin M}\right].
\end{aligned}$$

Combining the Union bound and (18), yields

$$\begin{aligned}
Pr\left[\neg \mathcal{P}_k \mid F_{S,\ell}^{(i)} \wedge \Phi_{u,v}^{\notin M}\right] &= Pr\left[\neg Q_k^{u_i} \vee \neg Q_k^{v_i} \mid F_{S,\ell}^{(i)} \wedge \Phi_{u,v}^{\notin M}\right] \\
&\leq Pr\left[\neg Q_k^{u_i} \mid F_{S,\ell}^{(i)} \wedge \Phi_{u,v}^{\notin M}\right] + Pr\left[\neg Q_k^{v_i} \mid F_{S,\ell}^{(i)} \wedge \Phi_{u,v}^{\notin M}\right] \\
&\leq \frac{2}{k}.
\end{aligned}$$

Hence, using similar arguments as in the proof of Lemma C.4, we have

$$Pr\left[\bigwedge_{k=\lfloor m/d \rfloor + 1}^{\ell-1} \mathcal{P}_k \mid F_{S,\ell}^{(i)} \wedge \Phi_{u,v}^{\notin M}\right] \geq \prod_{k=\lfloor m/d \rfloor + 1}^{\ell-1} \left(1 - \frac{2}{k}\right) = \frac{\lfloor m/d \rfloor - 1}{(\ell - 1) - 1} \cdot \frac{\lfloor m/d \rfloor}{\ell - 1}.$$

Therefore, it holds that

$$Pr\left[E^{(\ell)}\right] \geq \frac{\lfloor m/d \rfloor - 1}{(\ell - 1) - 1} \cdot \frac{\lfloor m/d \rfloor}{\ell - 1} \cdot Pr\left[\Phi_{u,v}^{\notin M}\right].$$

$\square$

## E.3   Proof of Theorem 5.3

In this section, we analyze the expected contribution in Phase III. Our goal now is to lower bound the expression

$$\mathbb{E}\left[A_\ell\right] = \mathbb{E}\left[w(e^{(\ell)}) \mid E^{(\ell)}\right] \cdot Pr\left[E^{(\ell)}\right].$$

It is straightforward to verify that Lemma C.3 holds in the current setting and yields

$$\mathbb{E}\left[w(e^{(\ell)}) \mid E^{(\ell)}\right] \geq \frac{OPT}{m}.$$

Further, by Lemma E.3, it holds for every $\ell \in \{\lfloor m/d \rfloor + 1, \ldots, m\}$ that

$$
\begin{aligned}
\mathbb{E}\left[A_\ell\right] &= \mathbb{E}\left[w(e^{(\ell)}) \mid E^{(\ell)}\right] \cdot Pr\left[E^{(\ell)}\right] \\
&\geq \frac{OPT}{m} \cdot \frac{\lfloor m/d \rfloor - 1}{(\ell-1)-1} \cdot \frac{\lfloor m/d \rfloor}{\ell-1} \cdot Pr\left[\Phi_{u,v}^{\notin M}\right],
\end{aligned}
$$

and thus

$$\sum_{\ell=\lfloor m/d \rfloor + 1}^{m} \mathbb{E}\left[A_\ell\right] \geq Pr\left[\Phi_{u,v}^{\notin M}\right] \cdot \frac{OPT}{m} \sum_{\ell=\lfloor m/d \rfloor + 1}^{m} \frac{\lfloor m/d \rfloor - 1}{(\ell-1)-1} \cdot \frac{\lfloor m/d \rfloor}{\ell-1}.$$

Hence, by combining the first inequality in (22) and Claim E.2, yields

$$
\begin{aligned}
\mathbb{E}\left[\sum_{\ell=\lfloor m/d \rfloor + 1}^{m} A_\ell\right] &\geq \left(\frac{d}{c}\right)^2 (1-o(1)) \cdot \left(\frac{d-1}{d^2} - o(1)\right) \cdot OPT \\
&\geq \left(\frac{d-1}{c^2} - o(1)\right) \cdot OPT.
\end{aligned}
$$

# F    Truthful mechanism for unit-demand domain

We first re-interpret some of the terminology in Section 4.[12] We have a set $L$ of agents and a set $R$ of items. Every agent $i \in L$ has a *(private) value* $v_i \geq 0$ for a set of preferred items $R_i \subseteq R$. We write $\Omega$ for the set of all (partial) matchings between $R$ and $L$, called *outcomes*, and $A_i$ for the set of all (partial) matchings in which $i$ gets matched up with some node in $R_i$, i.e., *satisfying outcomes* for agent $i$. We use $X_i : \Omega \to \{0, 1\}$ as the indicator function for the set $A_i$, i.e., we have $X_i(\omega) = 1$ if and only if $\omega \in A_i$. An agent's *type* is her value $v_i$, which is private information.

In the online setting, the agents arrive in a uniformly random order, and, upon arrival, an agent announces a (common) value $v_i'$ that she has for the items in $R_i$. The mechanism then commits to either choosing an outcome $\omega \in A_i$, that (partially) matches up all agents arrived so far, or some outcome $\omega \notin A_i$ that also (partially) matches up all agents arrived so far but not $i$, by definition of $A_i$. It also sets a price $\rho_i$,[13] which depends on the choice of $\omega$. We assume that $\rho_i \in \mathbb{N}_0$. The outcome $\omega$ should be consistent with previous outcomes in the sense that the proposed matching in step $i-1$ is a submatching of that committed to in step $i$.[14] After all agents have arrived the mechanism commits to the final outcome $\omega$ in step $n$. Informally speaking, whenever an agent arrives, the mechanism either offers her an item $r \in R_i$ at some price $p_i$, or it offers her nothing.

The goal of an agent is to maximize her utility

$$u_i(\omega) = v_i X_i(\omega) - \rho_i,$$

and the goal of the mechanism designer is to maximize the *social welfare*

$$\sum_{i \in L} v_i X_i(\omega).$$

We will show that Algorithm 4 can be used to design a truthful (online) mechanism in the case of so-called *single-value unit-demand domains*.

We want to design a mechanism that is *truthful*, meaning that it is always in the best interest of an agent to report $v_i' = v_i$ for any declarations of the other agents and any arrival order of the agents. We do this by showing that the procedures in Phase II and III can be modified so that the assignment rules in Algorithm 3 become *monotone*, see, e.g., Chapter 16 in [37]. It is not hard to see, and well-known, that this kind of monotonicity incentivizes truthfulness.

In order to turn Algorithm 3 into a truthful mechanism we need to make some modifications, and define a pricing scheme. The outcome $\omega$ in every step corresponds to the (online) matching $M$ we have chosen so far, together with the edge $e^i$ in case we match up $i$ when she arrives.

The modification for Phase I is straightforward. For every arriving agent $i$, we choose as outcome the empty matching (which is not in $A_i$ for any $i$), and set a price of $\rho_i = 0$. That is, we assign no item to any agent arriving in Phase I.

For Phase III we can use a similar idea as the *price sampling algorithm* in [4]. We set price-thresholds $\rho(r) = p_r^* - \lambda$. Whenever an agent arrives in Phase III, we look at all the items in $R_i$ for which her reported value $v_i'$ exceeds the price threshold $\rho(r)$ and assign her (if any) the item $r'$ for which the price threshold is the lowest among those. We charge her a price of $\rho_i = p_{r'}^* - \lambda$ for the item $r'$. The fact that this incentivizes truthful behavior follows from similar arguments as those given in [4].

In order to incentivize truthfulness in Phase II, we exploit the fact that we are free to choose any fixed algorithm to compute the (offline) matching $M^i$ in that phase once a node $i$ has arrived. That is,

Algorithm 3 works for every choice of such an offline algorithm, but in order to guarantee truthfulness, we need a specific type of bipartite matching algorithm.

We first introduce an additional graph-theoretical definition. We say that an instance of the bipartite matching problem on a (bipartite) graph $G = (A \cup B, E)$, with $|A| = n$, has uniform edge weights if all edges adjacent to a given node $a \in A$ have the same weight, i.e., we have $w(a, b) = w(a, b')$ for all $b, b' \in \mathcal{N}(a) \subseteq B$. We denote this common weight by $w_a$. Moreover, we write $(w'_a, w_{-a})$ to denote the vector $(w_1, \dots, w_{a-1}, w'_a, w_{a+1}, \dots, w_n)$ in which $w_a$ is replaced by $w'_a$.

**Definition F.1.** *We say that a deterministic (bipartite matching) algorithm $\mathcal{A}$ is* monotone *for instances with uniform edge weights if the following holds. For an instance $I = (G, (w_1, \dots, w_n))$ and any fixed $a \in A$, there exists a critical value $\tau_a = \tau_a(I)$ such that $\mathcal{A}$ has the following properties:*

1. *The node $a$ does not get matched for any $I = (G, (w'_a, w_{-a}))$ with $w'_a < \tau_a$;*

2. *There exists a node $b \in \mathcal{N}(a)$, such that for any $w'_a \geq \tau_a$ the node $a$ gets matched up to $b$ in $I = (G, (w'_a, w_{-a}))$.*

We emphasize that whenever $w'_a \geq \tau_a$, we want $a$ to be matched up to the *same* node $b$. So for any $w'_a$, the node $a$ either does not get matched up by $\mathcal{A}$, or it gets matched up to some fixed $b$.

Now, in Phase II we compute an offline matching $M^i$ using algorithm $\mathcal{A}$. If the edge $e^i = (i, r)$ assigned to $i$ in $M^i$ (if any) satisfies the condition that $M \cup e^i$ is a matching, where $M$ is the online matching constructed so far, we assign item $r$ to $i$. We charge a price of $\rho_i = \tau_a$. It is not hard to see, using a standard argument, that any monotone bipartite matching algorithm with the given pricing rule incentivizes truthful behaviour (this is left to the reader).

A summary of the modifications to Algorithm 3 can be found in Algorithm 7. Based on Algorithm 7, and the analysis of Algorithm 3, we obtain the following theorem, provided there exists a monotone bipartite matching algorithm $\mathcal{A}$ for instances with uniform edge weights. We will show the existence of such an algorithm in the proof of Theorem F.2.

**Theorem F.2.** *There is a deterministic truthful mechanism that is a $g_{\lambda,c,d}(\eta)$-approximation for the problem of social welfare maximization, with $g_{\lambda,c,d}$ as in Theorem 4.1. Furthermore, the alignment rule and pricing scheme of the mechanism are computable in polynomial time.*

*Proof.* As mentioned above, it suffices to show the existence of a monotone bipartite matching algorithm. Fix an arbitrary total order $\succ$ over all the elements of $L \times R$. We say that a matching $M = \{m_1, m_2 \dots\}$ is lexicographically larger than matching $M' = \{m'_1, m'_2, \dots\}$ with $M' \neq M$ and write $M \succ_{\text{lex}} M'$ if and only if, there exists an integer $k \geq 0$ such that $m_i = m'_i$ for $i = 1, \dots k$ and $m_{k+1} \succ m'_{k+1}$.

We claim that any existing exact algorithm for the maximum weight bipartite matching problem can be easily converted to an exact algorithm for the *lexicographically maximum weight bipartite matching problem*, i.e., the problem where one seeks to find lexicographically largest matching among the maximum weight matchings. Indeed, consider some existing algorithm $\mathcal{A}'$ for maximum weight bipartite matching, and assume that $\mathcal{A}'$ on instance $G = (L \cup R, E)$ gives a matching of cost OPT. Now consider the maximum $e = (\ell, r) \in E$ according to $\succ$, and compute the maximum weight bipartite matching in $G' = (\{L \setminus \{u\}\} \cup \{R \setminus \{v\}\}, E \setminus \{e' : e' \cap u \neq \emptyset \text{ or } e' \cap v \neq \emptyset\})$. If the resulting solution has cost $< \text{OPT} - w_e$, then we know that $e$ is not in any maximum weight bipartite matching, and can recursively continue with OPT and $G := (L \cup R, E \setminus \{e\})$. If on the other hand the resulting solution has cost $= \text{OPT} - w_e$, then $e$ is part of some maximum weight bipartite matching (and in particular the lexicographically maximal one), so we fix $e$ and compute the rest of the matching recursively with

---

**ALGORITHM 7:** Truthful online mechanism for single value unit-demand domains

---

**Input**   : Predictions $p^* = (p_1^*, \ldots, p_{|R|}^*)$, confidence parameter $\lambda > 0$, and $c > d \geq 1$. Monotone bipartite matching algorithm $\mathcal{A}$.

**Output:** Matching (or assignment) $M$ and prices $\rho = (\rho_1, \ldots, \rho_n)$.

**Phase I:**
**for** $i = 1, \ldots, \lfloor n/c \rfloor$ **do**
 | Assign no item to agent $i$ and set $\rho_i = 0$.
**end**
Let $L' = \{\ell_1, \ldots, \ell_{\lfloor n/c \rfloor}\}$ and $M = \emptyset$.
**Phase II:**
**for** $i = \lfloor n/c \rfloor + 1, \ldots, \lfloor n/d \rfloor$ **do**
  Let $L' = L' \cup \{i\}$.
  Let $M^i =$ optimal matching on $G[L' \cup R]$ computed using $\mathcal{A}$.
  Let $e^i = (i, r)$ be the edge assigned to $i$ in $M^i$.
  **if** $M \cup e^i$ *is a matching* **then**
    | Set $M = M \cup \{e^i\}$, i.e., assign item $r$ to $i$.
    | Set $\rho_i = \tau_i(G[L' \cup R], (v_1', \ldots, v_i'))$.
  **end**
**end**
**Phase III:**
**for** $i = \lfloor n/d \rfloor + 1, \ldots, n$ **do**
  Let $S = \{r \in \mathcal{N}(i) : v_i' \geq p_r^* - \lambda \text{ and } r \notin R[M]\}$
  **if** $S \neq \emptyset$ **then**
    | Set $M = M \cup \{i, r'\}$ where $r' = \text{argmax}\{v_i' - (p_r^* - \lambda) : r \in S\}$, i.e., assign item $r'$ to agent $i$.
    | Set $\rho_i = p_{r'}^* - \lambda$.
  **end**
**end**

---

$\mathrm{OPT} := \mathrm{OPT} - w_e$ and $G := G'$. By construction, we will in the end obtain the lexicographically largest maximum weight bipartite matching. We note that the lexicographically maximum weighted bipartite matching is unique for any input instance. Furthermore, we run algorithm $\mathcal{A}'$ at most $O(n^2)$ times. In other words we can determine such a matching in polynomial time for any given set of weights.

It remains therefore to show that the value $\tau_a$, as described in Definition F.1, exists. It is easy to see that there exists a value $w'_a$ for which $a$ is matched in any lexicographically maximum weighted bipartite matching (for example, set $w'_a = \sum_{e \in E \setminus a \cap E} w_e + \epsilon =: W$). Therefore it suffices to show that if $a$ is matched to some vertex $b \in \mathcal{N}(a)$ for some weight $w'_a = \tau_a$, then it is also matched to the same neighbor $b$ for any weight $w''_a > \tau_a$. Assume for the sake of contradiction that this is not the case, and let the corresponding two lexicographically maximum weighted matchings be $M' \ni (a, b)$ and $M''$. Note that $M''$ could leave $a$ unmatched, or it could contain an edge adjacent to $a$ but not $(a, b)$. In the following we use value$(M)$ to describe the total weight of matching $M$, i.e., $\sum_{e \in M} w_e$. We also note that, since $M'$ is a lexicographically maximal weighted bipartite matching for $(w'_a, w_{-a})$, we have that either value$(M'(w'_a, w_{-a})) >$ value$(M''(w'_a, w_{-a}))$ or the two values are equal and $M'(w'_a, w_{-a}) \succ_{\mathrm{lex}} M''(w'_a, w_{-a})$.

For the second case, and since the lexicographic order does not depend on the edge weights, we also have $M'(w''_a, w_{-a}) \succ_{\mathrm{lex}} M''(w''_a, w_{-a})$. Therefore, for $M''$ to be a lexicographically maximal weighted bipartite matching for $(w''_a, w_{-a})$, it has to be that

$$\mathrm{value}(M''(w''_a, w_{-a})) > \mathrm{value}(M'(w''_a, w_{-a})).$$

However, this is not possible, since value$(M'(w''_a, w_{-a})) =$ value$(M'(w'_a, w_{-a})) + w''_a - w'_a$, and

$$\mathrm{value}(M''(w''_a, w_{-a})) \leq \mathrm{value}(M''(w'_a, w_{-a})) + w''_a - w'_a,$$

which is a contradiction. For the first case:

$$\begin{aligned}
\mathrm{value}(M'(w''_a, w_{-a})) &= \mathrm{value}(M'(w'_a, w_{-a})) + w''_a - w'_a \\
&> \mathrm{value}(M''(w'_a, w_{-a})) + w''_a - w'_a \\
&\geq \mathrm{value}(M''(w''_a, w_{-a})),
\end{aligned}$$

This gives a contradiction. This concludes the proof for the first statement of the theorem.

For the second part, first recall that we run some deterministic algorithm for the bipartite weighted maximum matching $O(n^2)$ many times. It remains to show that $\tau_a$ can be computed. We know that $\tau_a$ exists and is in the range $[0, W]$. Also recall that $\tau_a$ is the smallest value for $w'_a$ such that $a$ gets matched in a lexicographically maximal weighted bipartite matching for $(w'_a, w_{-a})$. We therefore can perform a binary search over $[0, W]$, by trying out if in the corresponding instance $a$ is part of the lexicographically maximum weighted bipartite matching. Since we assume weights to be integers, this will terminate after at most $\log W$ many steps. By using for example the Edmonds-Karp algorithm for $\mathcal{A}'$, our mechanism has a running time of $O(n^4 m^2 \log W)$ which is polynomial in the input size. $\quad\square$