[Reviews · NeurIPS 2020]

Review 1

Summary and Contributions: Recently there has been a spate of work in online algorithms combining traditional online algorithms with “machine learned” advice. In such problems, one has access to an exogenous prediction about the problem, and one hopes for best of both worlds guarantees of the form: “if the prediction is good, then I do really well (beating the worst-case benchmark), but if the prediction is bad, then I still do (approximately) at least as well as the worst-case benchmark”. This paper initiates the study of online algorithms with machine learning advice for the secretary problem, the online bipartite matching problem, and the graphic matroid secretary problem. In the secretary problem, you observe a stream of n real numbers that arrives in a uniformly random order. Your goal is to choose the largest element (or at least achieve a good approximation to the largest element). In the setting with advice, you are given a prediction p of the maximum value of all n real numbers. The error eta is defined as the absolute difference between p and the true maximum. Note that if p is correct (eta = 0) you can perfectly solve the secretary problem by just waiting for the item with weight p and accepting it. For the secretary problem, the authors provide a family of algorithms parametrized by a lambda >=0 and a c>=1 such that (1) if eta > lambda (lots of error), it is 1/ce-competitive, and (2) if eta <= lambda, it is g(c, lambda)-competitive for some g(c, lambda) that grows to 1 as c->infty and lambda -> 0. The algorithm for the secretary problem with advice works in the following fashion: the stream is split into three segments. In the first segment, we just observe all the elements. In the second segment, we set a threshold based on the max of our advice and what the no-advice algorithm would tell us to do. Finally, in the third segment, we set a threshold based off of what a no-advice algorithm would tell us to do (given what we’ve observed in the first two segments). The results for the online bipartite matching problem and graphic matroid secretary problem are analogous but more involved. (In addition, their techniques lead to new no-advice algorithms for both these problems with nice properties; see Strengths).

Strengths: - The secretary problem is one of the central problems in online learning; understanding how to design algorithms for it given external advice is a natural and important direction. The other two online problems studied in the paper are also very natural. - Authors develop algorithms which effectively achieve a “best-of-both-worlds” guarantee for the three online problems they study, achieving at least the (best known) no-advice competitive ratio for two of the three problems. - Their techniques lead to a new no-advice deterministic online algorithm for the graphic matroid secretary problem a new no-advice truthful mechanism for the online bipartite matching mechanism (in a specific constrained setting).

Weaknesses: - It would be nice if there was some general statement connecting all three settings. What property does the problem need for these sorts of algorithms to work? - Is there any sense in which you can show that the tradeoffs you find between competitive ratio for different values of eta are tight? (I suspect this might be very hard).

Correctness: I have not checked the proofs of the propositions carefully, but they appear correct/reasonable at a first glance.

Clarity: The paper was well-written and easy to understand.

Relation to Prior Work: This paper cites the relevant prior work. It is a significant improvement over prior work; to my knowledge this is one of the first papers (the first paper?) studying the secretary problem and variants with machine-learned advice.

Reproducibility: Yes

Additional Feedback: I enjoyed reading this paper a lot. I think it is a strong paper; incorporating external advice into online learning algorithms is an active and interesting area of research, and this paper represents significant contributions to this area. Questions: - In the graphic matroid secretary problem, the advice is the form of “for each vertex, the maximum edge-weight adjacent to that node”. Does this actually encode the optimal spanning tree? To me it seems like it encodes some subset of the spanning tree, but not enough to necessarily recover the spanning tree. In this sense it feels a little bit weird - Is there a “for any c>=1” missing from the statement of Theorem 1.2?


Review 2

Summary and Contributions: The paper considers secretary problems (and related problems) in the setting where one seeks to maximize the expected value of the outcome and one is given a prediction as to what the maximum value is.

Strengths: The model and approach appear sensible. The results are fairly clearly written and make sense. The work appears novel, and in general these kinds of problems are of relevance to the NeurIPS community. The rapidly growing area of "algorithms with predictions" is nicely added to.

Weaknesses: The results are fairly straightforward if well executed. The role of prediction offers only slight variations on the standard analysis here. Their strategy is not shown to be optimal and because of its simplicity is suggests it can be improved on readily in future work.

Correctness: The work (proofs) appears to be correct.

Clarity: Generally the paper is well written although by necessity a lot appears in the supplemental. It's not especially clear when reading through why Lambert's function is making an appearance; while space is at a premium, I'd suggest making that clearer if at all possible in the main paper.

Relation to Prior Work: The paper discusses prior work suitably.

Reproducibility: Yes

Additional Feedback: Perhaps my main reasons for not pushing harder for acceptance are the following: 1) the value maximization version of the secretary problem is generally "easier" than the version trying to maximize the probability of finding the highest; and in that sense, this corresponds to their analysis being fairly straightforward. (Though, given the generalizations to multiple problems, quite useful.) 2) They do not provide lower bounds; it's a bit unclear that the "3-phased" approach given (observe, try to get prediction, then do the best you can) is optimal, although it would be very interesting if it were! Overall this is nice work and I'd generally be in favor of acceptance. Post author-feedback: I don't think the feedback affected my overall picture of the paper significantly. I still think it is a reasonably good paper and my rating reflects that it is marginally above the acceptance threshold.


Review 3

Summary and Contributions: The paper studies three classical online problem settings, namely the standrad secretary problem, online bipartite matching and secretary with a graphic matroid constraint, using machine learned advice. In this framework, the input is still generated adversarially and revealed in piecewise fashion to the algorithm, that still has to make immediate and irrevocable decisions. However, the algorithm also has access to advice, which is some extra information (often it could be the optimal offline solution itself) which is however not always accurate. The assumption is that an ML system provided some insight about the instance, which also comes with a confidence parameter. Then, given the input, the advice and its confidence, the online algorithm needs to adjust accordingly so that when the advice is actually good, the performance is better than the worst case competitive ratio, but even if the advice is arbitrarily wrong the performance never strays too far away from the worst case. To illustrate by briefly stating the algorithm and result for the standard secretary: the advice is (or tries to be) equal to the highest value item. Along with it, come two confidence parameters \lambda and c, indicating roughly how close it is to the real value and how much the advice should be trusted. Then, rather than proceeding in the standard secretary fashion, which is to skip the first 1/e items and then pick the max, the algorithm has 3 phases: in the first it just observes the sequence, in the second it uses the advice as well as the observed values and in the third it only uses the observations. Adjusting the length of each phase gives a parameterized result that produces an approximation ratio better than 1/e, when the advice is good enough and the confidence parameters are set accordingly. For the bipartite matching and graphical matroid case the algorithms are similar modifications of the currently best known ones, using one extra phase to incorporate the advice and then falling back to the usual algorithm.

Strengths: 1) The paper is written at a very high mathematical standard. 2) The secretary problem (and its variants) are theoretically important (and practically to a lesser extent) so a study through the lens of the advice framework would be of interests to researchers in online algorithms and algorithmic game theory.

Weaknesses: 1) The algorithms (although rigorously analyzed) are somewhat obvious modifications of the best known ones from the online literature. 2) The bounds have o(1) terms and start improving over the previously known results for arbitrarily long inputs. I am not sure how large these inputs needs to be, but it seems that this would seriously limit the applications of this approach. 3) I am a bit skeptical about how the competitive ratio results are presented. Ideally, I would have liked a result like the meta-result (or similar tradeoff) using only \lambda (or only c) and the prediction error. Instead the actual statements contain both the \lambda and c confidence parameters. This could be ok, but I don't think that a good tradeoff is obtained for all combination of values \lambda and c. So, is \lambda chosen first and then c optimised accordingly, for the worst case value of the prediction error? Is it the other way around? Since OPT is unknown, just knowing \lambda seems to provide very little information regarding the accuracy of the prediction and I don't think it's obvious at all what the actual tradeoff is, given the current statements. From a theoretical standpoint there are advantages to this presentation, but in this case I would prefer something different.

Correctness: I didn't have time to thoroughly examine the proofs, almost all of which are in the supplementary material. But, a quick read seems to indicate that the paper is written to a high mathematical standard and the arguments (including the flow of lemmata and theorems) sound logical.

Clarity: The paper is very well written in the mathematical and bibliographic exposition, but the theorem statements are quite hard to parse in their current form. There is some qualitative intuition, but I would have liked a clearer punchline.

Relation to Prior Work: The related work is thoroughly discussed.

Reproducibility: Yes

Additional Feedback: After reading the rebuttal and discussing with the other reviewers, I remain unconvinced that the parameters of the algorithms can be feasibly chosen, thus I cannot improve my score.


Review 4

Summary and Contributions: This paper considers the secretary and similar online non-adversarial problems in the setting where a machine learning estimator can provide some form of advice (with some prediction error). This is a recent setting where the goal is to use this advice to improve the competitive ratio of the online algorithm in the case where the advice is accurate, while not being penalized too much (within a constant factor) when the advice is inaccurate. In the case of the secretary problem, there is a stream of n numbers arriving in random order and the goal is to pick one so to maximize (the expected value of) its ratio to the maximum number. The typical approach picks the first number larger than the first n/e observed numbers and gives the well-known competitive ratio of 1/e. Now, given a prediction p of the highest number OPT (with unknown prediction error |p - OPT|), the authors introduce a generalization of the secretary algorithm which depends on a user-provided confidence parameter \lambda (how doubtful we are about the prediction accuracy) and a penalty parameter c (in case the prediction is wrong, we tolerate a 1/(ce) competitive ratio). The algorithm works as follows: (i) observe n/f(c) numbers and let z be their maximum; (ii) for n/g(c) steps, pick any number larger than max(z, p-\lambda); (iii) otherwise, ignore the prediction p, and pick any number larger than the ones seen so far. When the prediction p is close to OPT and lambda is small, this algorithm will beat the 1/e competitive ratio, otherwise it will pay at most 1/(ce). Analogous approaches are applied to online maximum-weight bipartite matching (with edges coming in a uniformly random order) and the graphic matroid secretary problem (which is equivalent to maximum-weight forest). In my opinion the problems and setting are interesting but it is hard to understand the significance of the competitive ratio improvements, especially considering that now the user is in charge of picking two hyper-parameters. I would have liked to see an experimental section to see the impact of these hyper-parameters: for example given a certain (small) c, how much of an accurate prediction and high confidence is needed in order to improve the competitive ratio. Minor: - L260: "I removed the footnote..." I believe this paragraph was unintended. - It's worth pointing out that for your secretary algorithm, \lambda doesn't need to be set until phase 2, after having seen a reasonable amount of samples. - Can any lower bound be proved in this setting?

Strengths: - Theoretically sound - Interesting setting

Weaknesses: - No experimental section - Setting of hyper-parameters may be non-trivial.

Correctness: Results are sound to me.

Clarity: The paper is well written but I would have liked the authors to elaborate more the consequences and corollaries of their main theorems.

Relation to Prior Work: Related work are properly presented.

Reproducibility: Yes

Additional Feedback:

[Author Response · NeurIPS 2020]

Firstly, we would like to thank all the reviewers for their careful reading and valuable suggestions. We will next address some of the main concerns raised in the reviews, as well as answer some of the questions that were posed.

Although the analysis of the secretary problem augmented with predictions (from a technical perspective) is mostly there to provide a clean and illustrative example, we give, in our opinion, a non-trivial analysis of the augmented online bipartite matching and the graphic matroid secretary problem. For these problems, we design novel algorithms whose analysis requires non-trivial probabilistic arguments, and crucially relies on the specific order of the procedures carried out in the Phases II and III. Albeit, due to space limitation, we defer them to the supplementary material:

i) for the online bipartite matching problem – see pages 14-15;

ii) for the graphic matroid secretary problem – see pages 19-20 and Appendix E.

This will be emphasized more in the paper.

We, indeed, do not provide any lower bounds for our algorithms, apart from the trivial lower bound induced by the original secretary problem. We agree that finding lower bounds is an important question (and based on personal experience, a difficult one) that would be very interesting to be addressed in future work. Further, finding a more general meta-property under which our results hold, would be very intriguing.

We also want to mention that $\lambda$ and $c$ in the analysis of the secretary problem are independent parameters that provide the most general description of the competitive ratio. Here $\lambda$ is our confidence of the predictions and $c$ describes how much we are willing to lose in the worst case. Although these parameters can be set independently, some combinations of them are indeed not very sensible, as one might not get an improved performance guarantee, even when the prediction error is small (for instance, if $c = 1$, i.e., we are not willing to lose anything in the worst case, then it is not helpful to consider the prediction at all). In particular, as one does not know the prediction error, there is no way in choosing these parameters optimally, since different prediction errors require different settings of $\lambda$ and $c$.

For the graphic matroid secretary problem, the predictions model the optimal spanning tree in the case when all edge-weights are pairwise distinct. Otherwise, there can be many (offline) optimal spanning trees, and thus the predictions do not encode a unique optimal spanning tree. We intentionally chose not to use predictions regarding what edges are part of an optimal solution, as in our opinion, such an assumption would be too strong.

With respect to the Lambert-W function, it appears naturally in the analysis of the classical secretary problem. In the classical analysis, 1/e is exactly the optimum solution where the two branches of the Lambert-W function "meet". We discuss this in more detail in the Preliminaries section and the proof sketch of Theorem 1.2.

There is indeed a missing expression "for any $c \geq 1$" in the statement of Theorem 1.2. Thank you for pointing this out.



[Meta-Review · NeurIPS 2020]

The author study the secretary problem and its extension in the paradigm of algorithms with predictions. The reviewers urge the authors to improve the discussion on how to set c and \lambda, particularly given their relationship with each other.